# Single-molecule electron spin resonance by means of atomic force microscopy

Lisanne Sellies[1 ✉], Raffael Spachtholz[1], Sonja Bleher[1], Jakob Eckrich[1], Philipp Scheuerer[1] & Jascha Repp[1 ✉]

Understanding and controlling decoherence in open quantum systems is of fundamental interest in science, whereas achieving long coherence times is critical for quantum information processing[1]. Although great progress was made for individual systems, and electron spin resonance (ESR) of single spins with nanoscale resolution has been demonstrated[2–4], the understanding of decoherence in many complex solid-state quantum systems requires ultimately controlling the environment down to atomic scales, as potentially enabled by scanning probe microscopy with its atomic and molecular characterization and manipulation capabilities. Consequently, the recent implementation of ESR in scanning tunnelling microscopy[5–8] represents a milestone towards this goal and was quickly followed by the demonstration of coherent oscillations[9,10] and access to nuclear spins[11] with real-space atomic resolution. Atomic manipulation even fuelled the ambition to realize the first artificial atomic-scale quantum devices[12]. However, the current-based sensing inherent to this method limits coherence times[12,13]. Here we demonstrate pump–probe ESR atomic force microscopy (AFM) detection of electron spin transitions between non-equilibrium triplet states of individual pentacene molecules. Spectra of these transitions exhibit sub-nanoelectronvolt spectral resolution, allowing local discrimination of molecules that only differ in their isotopic configuration. Furthermore, the electron spins can be coherently manipulated over tens of microseconds. We anticipate that single-molecule ESR-AFM can be combined with atomic manipulation and characterization and thereby paves the way to learn about the atomistic origins of decoherence in atomically well-defined quantum elements and for fundamental quantum-sensing experiments.

The experimental set-up is shown in Fig. 1a. Individual pentacene molecules were adsorbed onto a dedicated support structure to electrically gate the molecule against the tip potential and—at the same time—apply radio-frequency (RF) magnetic fields. This is achieved by a gold microstrip on a mica disc, covered by an insulating NaCl film that is thick enough to prevent electron tunnelling between the molecule and the microstrip. A gate voltage $V_G$ was applied to the microstrip to control single-electron tunnelling between the molecule and the conductive tip of the atomic force microscope[14]. RF magnetic fields were generated from an RF current sent through the microstrip. Experiments were performed at a temperature of 8 K.

To drive and probe ESR transitions, we first bring the closed-shell pentacene molecule to the excited triplet state $T_1$ by driving two tunnelling events with pump pulses of $V_G$ (refs. 15,16), first extracting an electron from the highest occupied molecular orbital (HOMO) and then injecting an electron into the lowest unoccupied molecular orbital (LUMO). The two unpaired electrons in the HOMO and LUMO form the triplet. As shown previously[15], the subsequent decay from $T_1$ into the singlet ground state $S_0$ can be measured by transferring the remaining population in $T_1$ after a controlled dwell time $t_D$ to the cationic charge state,

whereas pentacene in $S_0$ remains neutral (see Extended Data Fig. 1). The two charge states, cationic and neutral, can then be discriminated in the AFM signal[17] owing to their different electrostatic force acting on the tip during a probe period, allowing the population decay of state $T_1$ to be measured as a function of $t_D$ (ref. 15). Notably, the cationic state is only used to create the triplet state and facilitate the readout, whereas the ESR spectroscopy and spin manipulation described below occurs in the neutral triplet state.

As seen in Fig. 1b (red curve), the population decay of the $T_1$ state reflects the three lifetimes $\tau_X = 21$ μs, $\tau_Y = 67$ μs and $\tau_Z = 136$ μs of the three zero-field-split triplet states $T_X$, $T_Y$ and $T_Z$, differing from each other markedly (see Methods for the spin Hamiltonian). Driving an ESR transition between two of these states by an RF magnetic field of matching frequency effectively equilibrates their populations and thereby strongly affects the overall population decay of the $T_1$ state[18,19]. This is shown in Fig. 1b (black curve), for which the $T_X$–$T_Z$ transition was driven at 1,540 MHz. Around a $t_D$ of 100 μs, the change in the triplet population owing to the RF field is maximal for this transition.

To measure an ESR-AFM spectrum of this transition, we therefore fixed $t_D$ to 100.2 μs and recorded the triplet population as a function of

[1]Institute of Experimental and Applied Physics, University of Regensburg, Regensburg, Germany. ✉e-mail: lisanne.sellies@ur.de; jascha.repp@ur.de

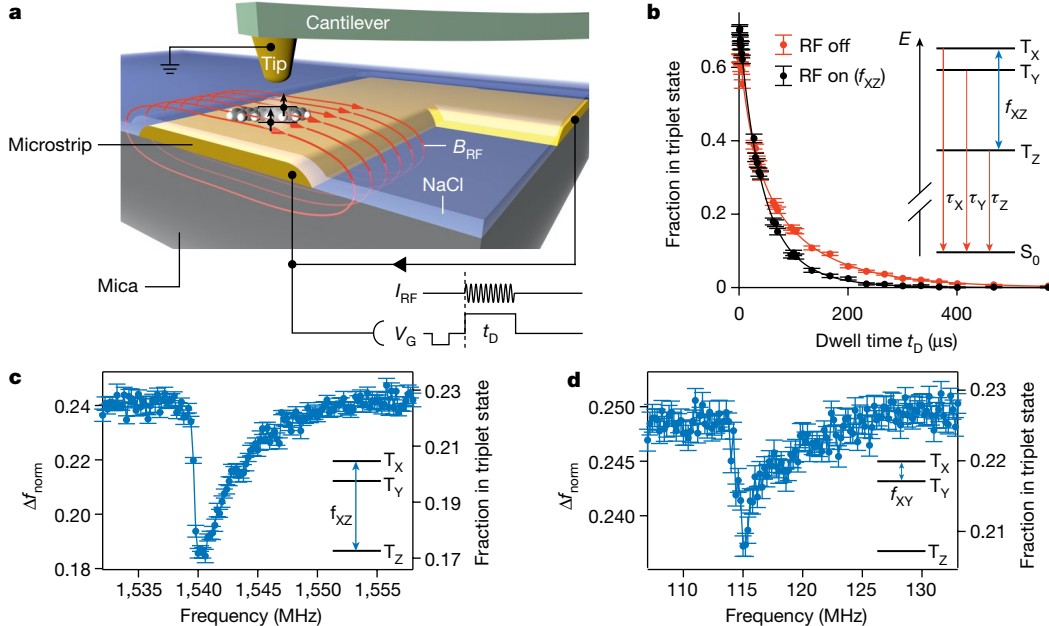

**Fig. 1 | Set-up, triplet decay under resonant driving and ESR-AFM spectra. a**, Sketch of the experimental set-up. Individual pentacene molecules were adsorbed on a Au(111) microstrip on a mica disc, covered by a NaCl film (>20 monolayers), preventing electron tunnelling between microstrip and molecule. A time-dependent gate voltage $V_G$ was applied to the strip to repeatedly bring the molecule in the neutral triplet excited state $T_1$ (represented by the two arrows) by two subsequent tunnelling events between molecule and conductive tip. During an experimentally controlled dwell time $t_D$, the neutral molecule can decay to the singlet ground state. An RF current $I_{RF}$ was run through the microstrip to generate an RF magnetic field. After $t_D$, the final state of the molecule was read out as described in Methods. **b**, Decay of the $T_1$ state as measured without RF (red) and with a broadband (see Methods) RF pulse (black). $T_1$ is zero-field-split into three states $T_X$, $T_Y$ and $T_Z$ having different lifetimes (inset), such that the RF pulse driving the $T_X$–$T_Z$ transition changes the resulting overall decay. Solid lines represent fits to triple-exponential decays. Each data point corresponds to 1,920 pump–probe cycles and the error bars were derived from the s.d.; see ref. 15. **c**,**d**, ESR-AFM spectra of the $T_X$–$T_Z$ and $T_X$–$T_Y$ transitions of a pentacene-$h_{14}$, respectively. The RF was swept at a constant $t_D$ = 100.2 µs. The AFM signal $\Delta f$ was normalized to $\Delta f_{norm}$ as described in Methods. It can be calibrated against the triplet population[15] at $t_D$; see right axes. The error bars were derived from the s.d. of seven and 38 measurements, respectively.

frequency $f_{RF}$ of the driving field. In this single-molecule experiment, the pump–probe cycle (Extended Data Fig. 1) was repeated 6,400 times in 20 s for every data point, whereas the AFM signal $\Delta f$ (see Methods) was measured and time averaged, yielding $\langle\Delta f\rangle$. This $\langle\Delta f\rangle$ is mainly determined by the probe phases because they make up for approximately 95% of the total time. As indicated above, during these probe phases, the outcome of the previous triplet-decay period is read out through the charge state of the molecule. From $\langle\Delta f\rangle$, a dimensionless normalized frequency shift $\Delta f_{norm}$ was derived, which scales linearly with the triplet population; for details, see Methods and Extended Data Fig. 2.

Figure 1c shows the resulting ESR signal with an asymmetric shape, which closely resembles the signal shape obtained with optically detected magnetic resonance (ODMR)[18,19]. This asymmetric shape entails information about the nuclear spin system of the molecule; it arises from the hyperfine coupling of the 14 proton nuclear spins to the electron spins (see Methods and Extended Data Fig. 3). Note that this signal was measured at low RF powers, at which power broadening[20] was negligible (Extended Data Fig. 4).

The aforementioned data-acquisition scheme was optimized for both swiftness and signal-to-noise ratio. We note that—at slower timescales—every single triplet-to-singlet decay event can be probed individually[15], providing absolute information about the spin population, as exemplarily demonstrated by the right axis in Fig. 1c.

Also, the $T_X$–$T_Y$ transition at 115 MHz can be probed with this technique (see Fig. 1d), being very similar to that for pentacene in a terphenyl matrix[21]. Note that the smaller ESR signal in this case is because of the smaller difference of the $T_X$ and $T_Y$ decay rates.

These ESR spectra were acquired in the absence of a static external magnetic field and therefore probe the zero-field splitting of $T_1$ (refs. 18,19). This splitting arises predominantly from the dipole–dipole

interaction of the two unpaired electron spins. It is therefore governed by the spatial distribution of the triplet state (see Extended Data Fig. 3 and Methods) and can serve as a fingerprint of the molecule. The shift of the $T_X$–$T_Z$ transition frequency by 60 MHz (about 4% of its value) with respect to pentacene molecules in a host matrix[18,19] can be rationalized by the different environments. Figure 2 demonstrates such fingerprinting for perylenetetracarboxylic dianhydride (PTCDA) in comparison with pentacene, along with an atomically resolved AFM image.

ESR-AFM, as introduced here, relies on selective triplet formation by electron tunnelling between the tip and the frontier orbitals of the molecule[15]. Therefore, the spatial resolution of the ESR-AFM signals is predominantly determined by the distance dependence of these tunnelling processes[22], allowing the unambiguous assignment of a given ESR spectrum to the individual molecule beneath the tip (see Extended Data Fig. 5). Similarly, the required tunnelling coupling restricts tip heights to a certain range (see Methods).

The narrow ESR line in Fig. 1c already indicates a long coherence time. To demonstrate the long coherence enabled by our new detection scheme, we measured Rabi oscillations[23]. The Rabi oscillations measured on a pentacene molecule shown in Fig. 3a demonstrate that coherent spin manipulation on a microsecond timescale is possible. In this experiment (Extended Data Fig. 6), the three triplet states were allowed to decay independently during $t_s$ = 45.1 µs, resulting in a strong imbalance of their populations, with the longest-lived $T_Z$ being dominant. Subsequently, an RF pulse of variable duration ($t_{RF}$ = 0 to 7 µs) at resonance with the $T_X$–$T_Z$ transition was applied, driving the population to oscillate between $T_Z$ and $T_X$. During the remaining roughly 50 µs of a fixed total $t_D$ = 100.2 µs, the triplet states again decayed independently from each other, such that—after each pulse sequence—predominantly the $T_Z$ population remained and was detected. Note

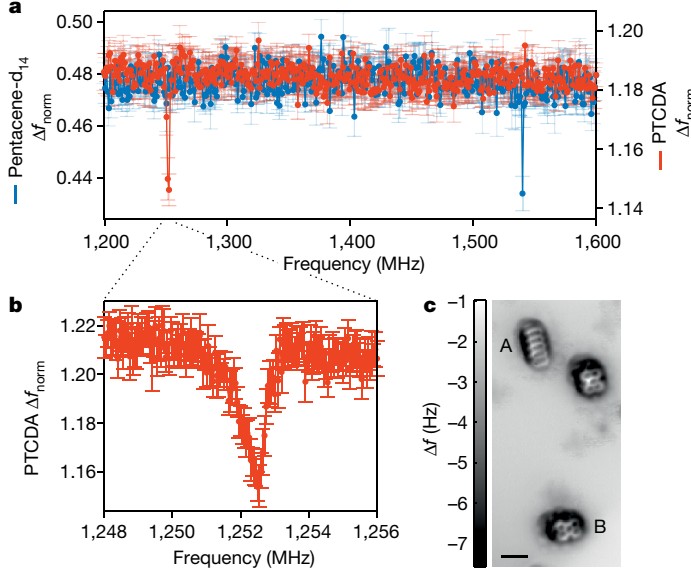

**Fig. 2 | ESR-AFM spectra of PTCDA and pentacene. a,** ESR-AFM spectra of pentacene-$d_{14}$ and PTCDA over a broad frequency range. Each molecule shows a sharp line at a characteristic frequency corresponding to its intrinsic zero-field splitting. Note that, for PTCDA, there is a very small signal at 1,501 MHz (not shown) that is too small to be observed at the parameters used here (see Methods). The molecules were measured under identical parameters (error bars are s.d. of twelve repetitions for PTCDA and seven repetitions for pentacene-$d_{14}$), except for $V_{deg}$ (which is specific to the molecule) and the dwell pulse parameters: $V_D = 2.5$ V, $t_D = 100.2$ μs for pentacene-$d_{14}$, $V_D = 2.35$ V, $t_D = 501$ μs for PTCDA (see Methods). **b,** Spectral zoom-in of the feature around 1,250 MHz of the PTCDA molecule, revealing its asymmetric lineshape (error bars are s.d. of eight repetitions). Note that the asymmetric shoulder appears here at the low-frequency side, indicative of the $T_Y$–$T_Z$ transition (see Methods). **c,** Constant-height AFM image atomically resolving two PTCDA and a pentacene molecule, measured with a CO-functionalized tip ($z$-offset $\Delta z = -3.1$ Å with respect to the set point: $\Delta f = -1.05$ Hz at $V = 0$ V, oscillation amplitude $A = 0.55$ Å). The two molecules, above which the spectra were taken, are labelled as 'A' for pentacene-$d_{14}$ and 'B' for PTCDA. The weaker features seen around the molecules are because of probe-tip imperfections. Scale bar, 10 Å.

that this Rabi-oscillation measurement scheme gives rise to an overall decaying trend of $\Delta f_{norm}$ (see blue line in Fig. 3a and Methods). The assignment as Rabi oscillations is confirmed by the linear dependence of the Rabi frequency on the RF amplitude (see Extended Data Fig. 7).

A decay constant of $2.2 \pm 0.3$ μs was extracted from the fit in Fig. 3a. Even though a single-molecule experiment avoids ensemble averaging, it still averages over possible fluctuations occurring in time. Here the nuclear spin configurations will fluctuate from pump–probe cycle

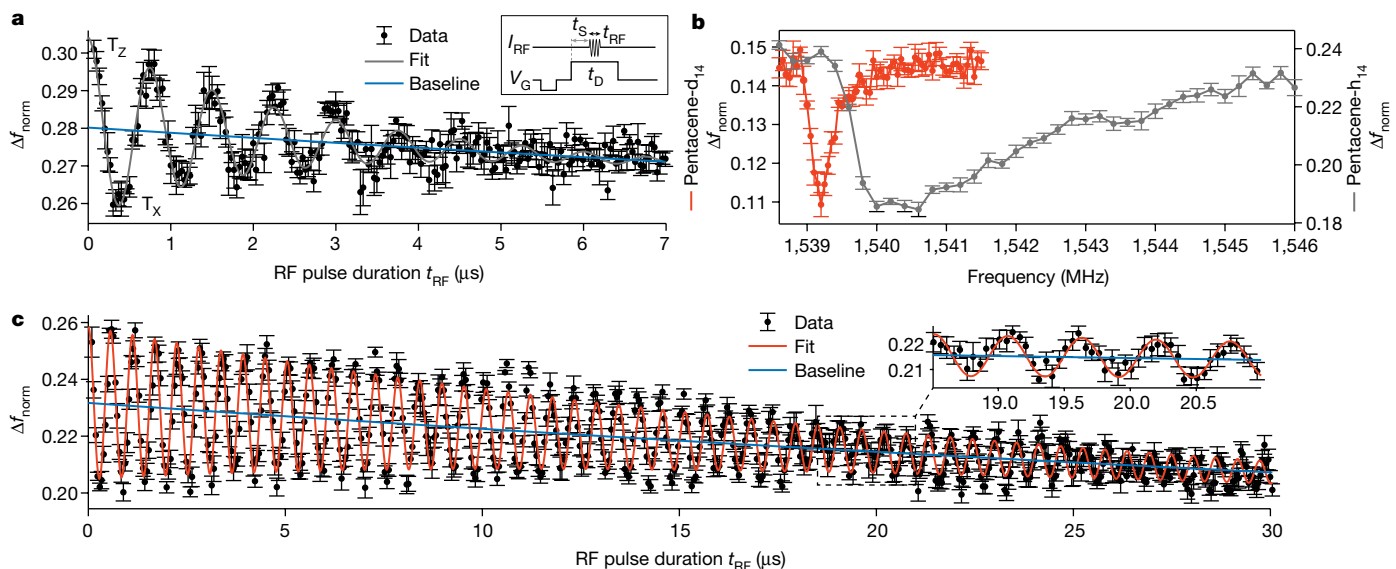

**Fig. 3 | Rabi oscillations and ESR-AFM spectra of a protonated and perdeuterated pentacene molecule. a,** Rabi oscillations from driving the $T_X$–$T_Z$ transition ($f_{RF} = 1,540.5$ MHz) showing coherent spin manipulation. The pump–probe pulse scheme is shown in the inset ($t_D = 100.2$ μs, $t_S = 45.1$ μs, $t_{RF}$ variable) and described in the main text (error bars are s.d. of four repetitions). The predominant contribution to $T_1$ is $T_Z$ at $t_{RF} = 0$, then starting to oscillate towards a predominant contribution of $T_X$ and back, as indicated for the first oscillation. A fit (grey line) yields a decay constant of the Rabi oscillations of $2.2 \pm 0.3$ μs (see Methods for details on the fit of the baseline (blue line)).

**b,** ESR-AFM spectrum of pentacene-$d_{14}$ (red), exhibiting a much narrower resonance in comparison with pentacene-$h_{14}$ (grey). The decreased hyperfine interaction leads to a reduced width of the high-frequency tail. The left flank of the signal corresponds to a broadening of only 0.12 MHz. The error bars result from the s.d. of eight (pentacene-$d_{14}$) and seven (pentacene-$h_{14}$) repetitions. **c,** The Rabi oscillations of pentacene-$d_{14}$ have a longer decay time of $16 \pm 4$ μs. The pump–probe pulse scheme was the same as that used for **a** (error bars are s.d. of eight repetitions) but with $t_S = 30$ μs (see Methods).

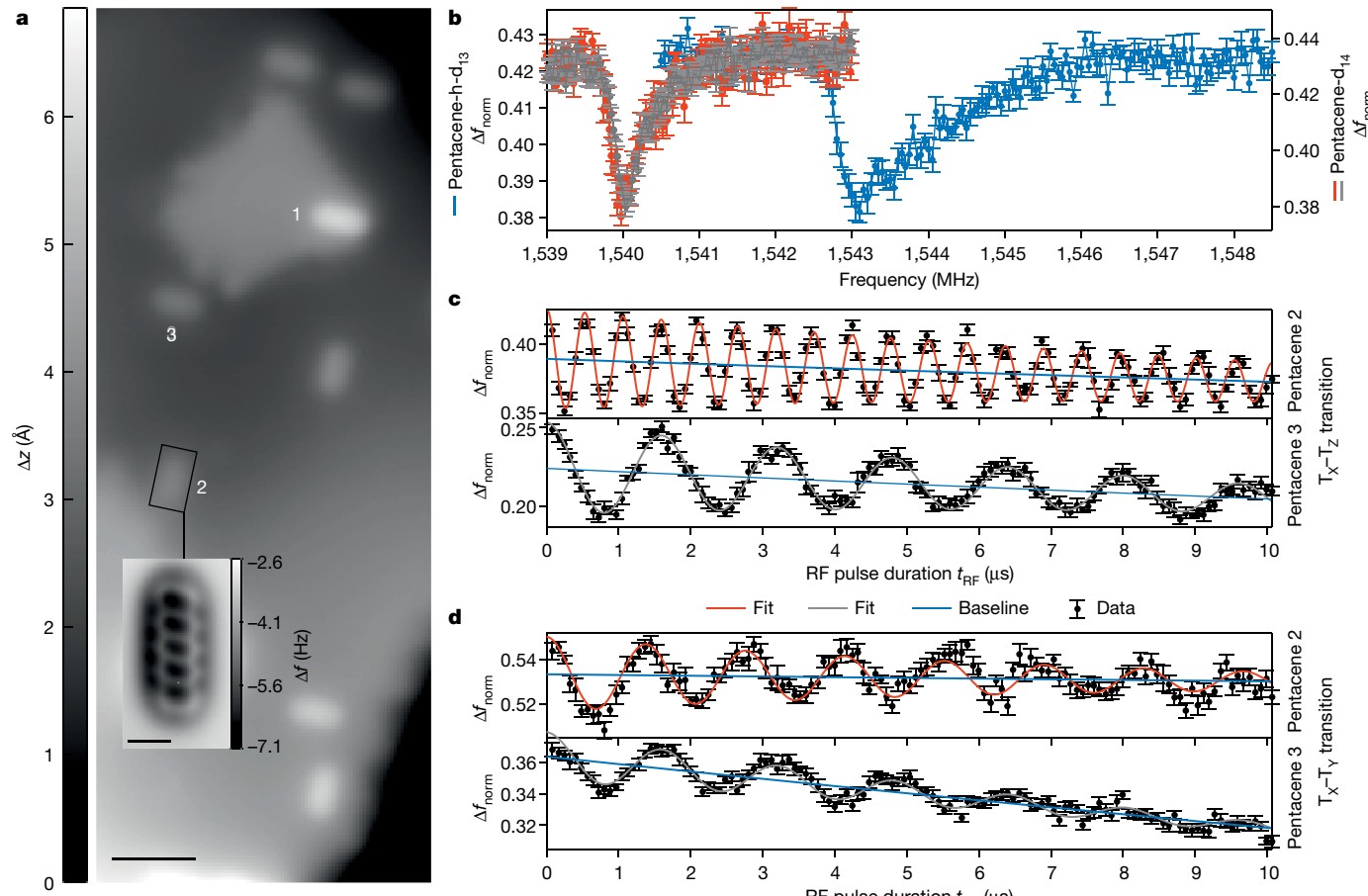

**Fig. 4 | ESR-AFM spectra and Rabi oscillations of differently oriented molecules and different isotopologues of pentacene. a**, AFM topography image of the NaCl-covered surface with several individual pentacene molecules measured with a CO-functionalized tip (set point: $\Delta f = -1.45$ Hz at $V = 0$ V, oscillation amplitude $A = 1.65$ Å). The inset shows a constant-height AFM image as a close-up of molecule 2 resolving its structure ($A = 0.3$ Å, $\Delta z = -5.08$ Å with respect to the set point $\Delta f = -1.45$ Hz at $V = 0$ V, $A = 0.3$ Å). Scale bars, 30 Å (main), 5 Å (inset). **b**, Although the ESR-AFM spectra of two individual pentacene-$d_{14}$ (red (denoted '2' in **a**) and grey (denoted '3')) molecules are very similar, that of

another isotopologue, pentacene-h-$d_{13}$ (blue (denoted '1')) differs clearly (note that the grey dataset is offset by 0.19 with respect to the right axes) (error bars are s.d. of four repetitions). **c**,**d**, Rabi oscillations of the $T_X$–$T_Z$ and $T_X$–$T_Y$ transitions (error bars are s.d. of eight and ten repetitions, respectively) of the two individual molecules 2 and 3. Although for the same RF power the Rabi frequency of the $T_X$–$T_Y$ transition for the two molecules is comparable, that of the $T_X$–$T_Z$ transition differs by almost a factor of three, in agreement with their different adsorption orientation and the selection rules (see text). For the reproducibility on other individual molecules, see Extended Data Fig. 9.

to pump–probe cycle, giving rise to the peculiar lineshape reflecting all different nuclear spin configurations (Fig. 1c,d). Similarly, the measured oscillations represent an average over finding the individual pentacene molecule in different nuclear spin configurations and, consequently, of a resonance and thus a Rabi frequency differing for every individual pump–probe cycle. Hence, the observed decay of the oscillations is probably dominated by the dephasing from the fluctuating Rabi frequency, limiting the coherence time[24]. Reducing the hyperfine interaction could then increase the coherence time further.

To this end, we studied pentacene-$d_{14}$, that is, fully deuterated pentacene, the ESR spectrum of which is shown in Fig. 3b. Comparing with pentacene-$h_{14}$, the peak shape is similar but its high-frequency tail is reduced in width by a factor of approximately 14. We suggest this reduction to be the result of the smaller hyperfine interaction of deuterium; the width is now probably dominated by nuclear electric quadrupole interaction[21]. The left flank of the signal exhibits a broadening of 0.12 MHz (full width at half maximum; see Methods), corresponding to less than a nanoelectronvolt in spectral resolution. The Rabi oscillations on pentacene-$d_{14}$ exhibit a much longer decay time of $16 \pm 4$ μs (see Fig. 3c). This shows that, in the above-described experiments on

protonated pentacene, the coherence was limited by the molecule itself and not by the method.

Notably, in force-detected ESR, as introduced here, no decoherence owing to tunnelling electrons is induced, as not even a single electron needs to be sent through the molecule during the ESR pulse. Further, as ESR-AFM does not rely on a finite conductance to the substrate, scattering with the electron bath of a conducting substrate is absent as a decoherence source[9,12,25] and there are no electronic states available for scattering in the thick NaCl film close to the Fermi level. Although scattering with conduction electrons in the tip remains possible, decoherence owing to the latter is expected to be small because of the weak tunnel coupling. Indeed, varying the tip height and cantilever oscillation amplitude does not appreciably affect the Rabi oscillations (see Extended Data Fig. 8). Moreover, the ESR-AFM technique introduced here does not require a magnetic tip[2] and thereby avoids interaction between the spin system and the magnetic stray field of the tip[5,9,13]. Nonetheless, the molecules under study are subject to many further interactions and decoherence sources, such as hyperfine coupling inside the molecule[24] and to nuclei in the substrate, coupling to neighbouring molecules[26] and to substrate defects such as step edges, and the electric field penetrating from the tip. We believe that, by avoiding

various strong decoherence sources, ESR-AFM will give access to these interactions of the environment occurring on a much smaller magnitude of coupling. For example, we observed an appreciable Stark shift of the $T_X$–$T_Z$ transition[27] of pentacene-$d_{14}$ (not shown) on the order of 0.3 MHz on changing the voltage by 1 V in the tip–sample junction. It seems likely that this Stark shift together with the cantilever oscillation contributes to the small but finite linewidth, as well as the observed decoherence in our experiments. The Stark shift might also be exploited in future scanning-gate-type experiments[28].

Demonstrating the combination of single-spin sensitivity and atomic-scale local information, we locally identified a single pentacene-h-$d_{13}$ of the otherwise deuterated pentacene molecules from its spectral signature (comparing with ODMR data[29]) and imaged it in its unique environment, as shown in Fig. 4a,b. The hyperfine interaction generally offers a way to manipulate and probe nuclear spins—featuring even longer coherence times[30]—by means of the electronic spin system[31]. Further, the interplay of selection rules and molecular orientation can be visualized at atomic scales, as shown in Fig. 4c,d: the $T_X$–$T_Z$ transition is driven by the component of the RF magnetic field along the short molecular axis[32] (see Methods), which obviously depends on the azimuthal orientation of the molecule, giving rise to a corresponding Rabi frequency being proportional to the projection of the RF field along the short axis of the molecule (see Fig. 4c). By contrast, the $T_X$–$T_Y$ transition is driven only by the RF-field component perpendicular to the molecular plane. As the latter always coincides with the surface plane, the corresponding Rabi frequency should be the same for all molecules, as exemplified in Fig. 4d.

The single-molecule ESR-AFM as introduced here opens several new research directions simultaneously. The system is brought to and studied at out-of-thermal-equilibrium and thereby eliminates the need to measure at (below) liquid-helium temperatures and large magnetic fields. Experiments in the absence of a static external magnetic field enable fingerprinting molecules from their zero-field splitting. The spin coherence on the 10-μs scale demonstrated here represents a leap forward for local studies of future artificial quantum systems and fundamental local quantum-sensing experiments. Directly resolving the atomic-scale geometry in relation to minute differences in zero-field splitting, spin–spin coupling, spin decoherence, as well as hyperfine coupling will boost our atomistic understanding of the underlying mechanisms. In combination with the toolset of atom manipulation, such as spin[33] and charge-state control[17], this will open a new arena for future fundamental studies.

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

# Methods

## Set-up and sample preparation

Our measurements were performed under ultrahigh vacuum (base pressure, $p < 10^{-10}$ mbar) with a home-built conductive-tip atomic force microscope equipped with a qPlus sensor[34] (resonance frequency, $f_0 = 30.0$ kHz; spring constant, $k \approx 1.8$ kNm$^{-1}$; quality factor, $Q \approx 1.9 \times 10^4$ and $2.8 \times 10^4$) and a conductive Pt-Ir tip. The microscope was operated in frequency-modulation mode, in which the frequency shift $\Delta f$ of the cantilever resonance is measured. The cantilever amplitude was 0.55 Å (1.1 Å peak to peak), except if specified otherwise. Constant-height AFM images were taken at tip-height changes $\Delta z$ with respect to the set point, as indicated. Positive $\Delta z$ values indicate being further away from the surface.

As a sample substrate, we used a cleaved mica disc, on which we deposited gold in a loop structure (diameter, $d = 10.5$ mm; thickness, $t = 300$ nm) by means of electron-beam physical vapour deposition. This gold structure contained a 100-µm-wide constriction, on which the measurements were performed. A non-conducting spacer material was introduced below the mica disc to prevent eddy-current screening of the RF magnetic field. The sample was prepared by short sputtering and annealing cycles (annealing temperature, $T \approx 550$ °C) to obtain a clean Au(111) surface. On half of the sample, a thick NaCl film (>20 monolayers) was grown at a sample temperature of approximately 50 °C; the other half of the sample was used for tip preparation, presumably resulting in the tip apex being covered with gold. Part of the data was measured with a CO-functionalized tip apex. To this end, a sub-monolayer coverage of NaCl was also deposited on the whole surface at a sample temperature of approximately 35 °C, to grow two monolayer NaCl islands also on the half of the sample used for tip preparation. After preparing a tip by indenting into the remaining gold surface, a CO molecule was picked up from the two monolayer NaCl islands, after which the tip was transferred to the thick NaCl film[35]. The NaCl film inhibits any electrons to tunnel to or from the gold structure. The voltage that is applied to the gold structure with respect to the tip represents a gate voltage ($V_G$), gating the molecular electronic states against the chemical potential of the conductive tip. The measured molecules (pentacene-h$_{14}$ and PTCDA-h$_8$, Sigma-Aldrich; pentacene-d$_{14}$, Toronto Research Chemicals) and CO for tip functionalization were deposited in situ onto the sample inside the scan head at a temperature of approximately 8 K. Pentacene was reported to adsorb centred above a Cl$^-$ anion with the long molecular axis aligned with the polar direction of NaCl, resulting in two equivalent azimuthal orientations[36].

The AC voltage pulses were generated by an arbitrary waveform generator (TGA12104, Aim-TTi), combined with the DC voltage, fed to the microscope head by a semi-rigid coaxial high-frequency cable (Coax Japan Co. Ltd.) and applied to the gold structure as $V_G$. The high-frequency components of the pulses of $V_G$ lead to spikes in the AFM signal because of the capacitive coupling between the sample and the sensor. To suppress these spikes, we applied the same pulses with opposite polarity and adjustable magnitude to an electrode that capacitively couples to the sensor.

The RF signal was produced by a software-defined radio (bladeRF 2.0 micro xA4, Nuand), low-pass filtered to eliminate higher-frequency components and amplified in two steps (ZX60-P103LN+, Mini Circuits; KU PA BB 005250-2 A, Kuhne electronic). The RF was pulsed using RF switches (HMC190BMS8, Analog Devices), which were triggered by the arbitrary waveform generator, allowing synchronization with $V_G$ and control over the pulse duration. The pulsed RF signal was fed into the microscope head by a semi-rigid coaxial high-frequency cable (Coax Japan Co. Ltd.) ending in a loop, inductively coupling the RF signal to the gold loop on the sample. These two loops are in the surface plane of the sample, such that the inductive coupling adds a vertical $z$ component to the magnetic field. The field generated by the microstrip is associated to field lines looping around the microstrip (Ampère's law).

At the position of a molecule placed above the strip, the local magnetic field resulting from the microstrip is expected to be homogeneous, in the surface plane and perpendicular to the direction of the microstrip.

The RF signal transmission of the cables including the loop for inductive coupling was detected by a magnetic field probe and can be well approximated to be constant over intervals of tens of megahertz around the $T_X$–$T_Z$ transition, that is, wider than the spectral features observed in the experiments. Although the microstrip will contribute to the overall transmission of the signal to the local magnetic field, it is expected to not introduce any resonances in the frequency range of interest. Note that the RF signal at a frequency of 1,500 MHz has a wavelength roughly three times the entire circumference of the loop of the microstrip.

To excite the entire broadened ESR resonance for the lifetime measurements of the triplet state $T_1$ with RF, shown in Fig. 1b, we used IQ modulation to generate a broadband RF pulse. We created a chirped pulse with a width of 12 MHz, a repetition time of 5 µs and a centre frequency of 1,544 MHz. Thereby, the RF signal spans the range 1,538–1,550 MHz in frequency space.

## ESR-AFM pulse sequence and data acquisition

The description of the measurement of the triplet-state lifetime can be found in ref. 15. The ESR-AFM experiments were performed with a similar voltage-pulse sequence, which is shown in Extended Data Fig. 1. Between each individual voltage-pulse sequence, the voltage is set to $V_{deg}$, the bias voltage, at which the respective ground states of the positively charged ($D_0$) and the neutral ($S_0$) molecules are degenerate. This way, the spin states of the molecule are converted to different charge states and detected[15] by charge-resolving AFM (ref. 17). The dwell voltage pulse duration $t_D$ was fixed to 100.2 µs for pentacene and, simultaneously, an RF pulse with a variable frequency was applied. In the case of PTCDA, the triplet lifetimes were determined to be $350 \pm 43$ µs, $170 \pm 13$ µs and $671 \pm 62$ µs, and $t_D$ was set to 501 µs for the ESR-AFM spectra shown. To reduce the statistical uncertainty for a given data-acquisition time, we repeated the pump–probe pulse sequence 160 or 320 times per second (instead of eight times per second for the lifetime measurements of the triplet state $T_1$). Note that, to prevent the excitation of the cantilever, the durations of the voltage pulses were set to an integer multiple of the cantilever period (33.4 µs). At this high repetition rate of the voltage-pulse sequence, the charge states cannot be read out individually. Instead, the AFM signal, that is, the frequency shift $\Delta f$, was averaged over an interval of 20 s. This average frequency shift $\langle \Delta f \rangle$ reflects the ratio of the charged and neutral states and thus the triplet and singlet states, but because the change in $\Delta f$ is very small, it is also sensitive to minor fluctuations in the tip–sample distance (see Extended Data Figs. 2c and 5b). To minimize the fluctuations in tip–sample distance, the tip–sample distance was reset by shortly turning on the $\Delta f$ feedback either after every sweep of the RF or after a fixed time (15 to 60 min). To minimize the dependence of $\langle \Delta f \rangle$ on the remaining fluctuations in tip height, $\langle \Delta f \rangle$ was normalized using the frequency shifts of the charged $\Delta f^+$ and neutral $\Delta f^0$ molecule, as $\Delta f_{norm} = \frac{\langle \Delta f \rangle - \Delta f^0}{\Delta f^+ - \Delta f^0}$. These frequency shifts were determined at the beginning and end of every 20-s data trace (see Extended Data Fig. 2a); the charge state was changed by applying small voltage pulses ($V_{set}^0 = V_{deg} + 0.3$ V, $V_{set}^+ = V_{deg} - 0.3$ V). Tunnelling events during the readout of these frequency shifts were minimized by using a tip–sample distance at which the decay constant for the decay of the $D_0$ state into the $S_0$ state during a pulse of $V_{deg} + 1.2$ V was around 4 µs (note that this requirement restricts the possible distances to a small range, as the tip height should also be small enough such that the tunnelling processes are considerably faster than the triplet decay; see also discussion of the spatial resolution in the main text).

If still a charging event happened, the data trace was discarded. To maximize the rate of the tunnelling processes during the voltage-pulse sequence, the beginning and end of the voltage pulses were synchronized with the closest turnaround point of the cantilever movement.

The data-acquisition and renormalization scheme to derive $\Delta f_{norm}$ is shown in Extended Data Fig. 2. As can be seen in Extended Data Figs. 2c and 5b, also the raw $\langle \Delta f \rangle$ signal exhibits the ESR features but with stronger baseline drift.

Note that $\Delta f_{norm}$ typically deviates from the triplet population, but that, for a given measurement, a linear relation between them exists. This deviation arises from the voltage pulses that are for pentacene turned on for 4.3% of the time, during which the frequency shift corresponds to the applied voltages and thus crucially depends on the exact shape of the Kelvin probe force parabola[17]. This explains the differences in the baseline of the $\Delta f_{norm}$ signal (without RF or RF off-resonance) for different measurements—even for those above the same molecule—owing to differences in the position above the molecule. Quantitative results (right axis of Fig. 1c,d) can be obtained from a calibration measurement in which the population was determined by counting the individual outcomes after each pulse sequence at a repetition rate of eight per second. This calibration was performed for an RF corresponding to the maximum of the ESR signal, as well as an RF that was off-resonance. For both cases, 7,680 pump–probe cycles were recorded.

Note that we do not observe any appreciable change in the damping of the cantilever during an RF sweep.

## Experimental uncertainties and statistical information

To determine the uncertainty on the ESR-AFM data points, the 20-s data traces were repeated several times and the error bars were extracted as the s.d. of the mean of these repetitions. This way, any type of non-systematic uncertainty will be accounted for, irrespective of its source (see next paragraph). Note that the hydrogen spins can have a different configuration for every individual readout[19]. Given our large number of sampling events, we acquire an average over the possible nuclear spin configurations.

The three main sources of uncertainty of $\Delta f_{norm}$ are the statistical uncertainty from the finite number of repeats[15], the remaining drift of the tip height and the noise on the frequency shift $\Delta f$. We choose the number of repeats per data point such that the statistical uncertainty becomes comparable with the other two sources of uncertainties; depending on the exact experimental conditions, any of these three sources can dominate. Probe tips that give a strong response to charging (a large charging step in the Kelvin parabola) provide a better signal-to-noise ratio and, therefore, a smaller relative uncertainty. To minimize this contribution to the uncertainty, we only used tips for which the charging step was large compared with the noise in $\Delta f$ (size of charging step 0.2–0.4 Hz for tips with a $\Delta f$ setpoint around −1.5 Hz at zero bias; $\Delta f$ averaged over 1 s exhibits a typical uncertainty of 1 mHz). In the case of the data in Fig. 4c, top and Extended Data Fig. 8c, bottom, the drift was clearly dominating the error margins. Therefore, the noise resulting from drift was, for these datasets, further minimized by setting the average of every repeat equal to the average over all data points.

In the case of pentacene, the $T_X$–$T_Z$ transition was measured for 19 individual pentacene-$h_{14}$ molecules, 20 pentacene-$d_{14}$ and 1 pentacene-$h$-$d_{13}$; for 18 of these molecules, we also measured the $T_X$–$T_Y$ transition. The $T_Y$–$T_Z$ transition of PTCDA was measured in 20 individual spectra for 2 molecules, whereas $T_X$–$T_Y$ and $T_X$–$T_Z$ transitions (not shown) were measured 3 and 5 times, respectively. In total, 16 different tips were used for these measurements. Molecule-to-molecule variations of the resonance frequencies for two different tips are shown in Extended Data Table 1.

## Spin Hamiltonian and eigenstates

The pulse sequence prepares the molecule into an electronically excited state with one unpaired electron in both the HOMO and the LUMO. These electrons couple through exchange interaction, leading to a large energy difference of $\Delta E \approx 1.1$ eV (refs. 37,38) between the excited singlet $S_1$ and the excited triplet $T_1$ states. We note in passing that this energy difference allows us to selectively occupy $T_1$ instead of $S_1$. With respect to exchange interaction, all three triplet sub-states of $T_1$ are degenerate. These are typically represented in the basis of magnetic quantum numbers $m_S = -1$, 0 and +1, which—in the representation of the two coupled spins—is $T_{-1} = |\downarrow\downarrow\rangle$, $T_0 = (|\uparrow\downarrow\rangle + |\downarrow\uparrow\rangle)/\sqrt{2}$ and $T_{+1} = |\uparrow\uparrow\rangle$, respectively.

As explained in the following, the magnetic dipole–dipole interaction between the two electron spins, which is orders of magnitude weaker than the exchange interaction, lifts this degeneracy, leading to a splitting of the triplet states, called the zero-field splitting. We note that the zero-field splitting may also have contributions from spin–orbit interaction. The dipole–dipole interaction is described by the Hamiltonian[39]

$$\mathscr{H} = -\frac{\mu_0 \gamma_e^2}{4\pi r^3}(3(\mathbf{S}_1 \cdot \hat{\mathbf{r}})(\mathbf{S}_2 \cdot \hat{\mathbf{r}}) - \mathbf{S}_1 \cdot \mathbf{S}_2)\hbar^2,$$

with the two spins $\mathbf{S}_1$ and $\mathbf{S}_2$ at a distance $r$ in a relative direction $\hat{\mathbf{r}} = \mathbf{r}/r$. $\mu_0$ is the magnetic constant, $\gamma_e$ the gyromagnetic ratio, $\hbar$ the reduced Planck constant and $\mathbf{r}$ the vector connecting the two spins. Notably, the magnetic dipole–dipole interaction is highly anisotropic, that is, for given spin orientations, it strongly differs and even changes sign for different relative positions of the two spins (see Extended Data Fig. 3a). The spatial positions of the electron spins are given by the orbital densities of the two electrons, the confinement of which is very different along the three molecular axes (see Extended Data Fig. 3b). Note that, for pentacene and PTCDA, the $z$ direction is perpendicular to the molecular plane and, thereby, perpendicular to the surface plane; $x$ points along the long molecular axis[21]. The anisotropy of the dipole–dipole interaction together with the non-uniformity of orbital densities gives rise to an energy difference in the range of microelectronvolts for the spins pointing in different real-space dimensions. This zero-field splitting is thus a fingerprint of the orbital densities and thereby the molecular species (see Fig. 2).

The corresponding eigenstates are no longer $T_{-1}$, $T_0$ and $T_{+1}$ but $T_X$, $T_Y$ and $T_Z$. The latter eigenstates expressed in the basis of the former read $T_X = (T_{-1} - T_{+1})/\sqrt{2}$, $T_Y = (T_{-1} + T_{+1})i/\sqrt{2}$ and $T_Z = T_0$, whereas expressed as the states of the two individual spins $|m_{s1} m_{s2}\rangle$, they are $T_X = (|\downarrow\downarrow\rangle - |\uparrow\uparrow\rangle)/\sqrt{2}$, $T_Y = (|\downarrow\downarrow\rangle + |\uparrow\uparrow\rangle)i/\sqrt{2}$ and $T_Z = (|\uparrow\downarrow\rangle + |\downarrow\uparrow\rangle)/\sqrt{2}$. Further, they have the property that the expectation value of the total spin $\langle T_i|\mathbf{S}|T_i\rangle$ vanishes for all three states $T_{i=X,Y,Z}$, whereas $\langle T_i|\mathbf{S}_j^2|T_i\rangle = (1 - \delta_{ij})$. Here $\delta_{ij}$ is 0 for $i \neq j$ and 1 for $i = j$. $\langle \mathbf{S} \rangle = 0$ renders these triplet states relatively insensitive to external perturbations; an external magnetic field affects the system and energies only to the second order.

The spin Hamiltonian $\mathscr{H}$ for the zero-field splitting and an external magnetic field $\mathbf{B}$ (excluding hyperfine terms) is $\mathscr{H} = \mathbf{S}\hat{D}\mathbf{S} + g_e\mu_B\mathbf{S}\mathbf{B}$, with the dipole–dipole-interaction tensor $\hat{D}$. Explicitly expressed in the basis of the zero-field split states $T_X$, $T_Y$ and $T_Z$, it reads[39]

$$\mathscr{H} = \begin{bmatrix} \epsilon_X & -ig_e\mu_B B_Z & ig_e\mu_B B_Y \\ ig_e\mu_B B_Z & \epsilon_Y & -ig_e\mu_B B_X \\ -ig_e\mu_B B_Y & ig_e\mu_B B_X & \epsilon_Z \end{bmatrix}$$

Here $\mu_B$ is the Bohr magneton, $g_e$ is the electron $g$-factor and $\epsilon_X$, $\epsilon_Y$ and $\epsilon_Z$ are the zero-field energies of $T_X$, $T_Y$ and $T_Z$, respectively. With increasing external magnetic field, the eigenstates will gradually change and asymptotically become the states $T_{-1}$, $T_0$ and $T_{+1}$ in the limit of large magnetic fields (for example, see Extended Data Fig. 3c).

## Selection rules

It follows from the above Hamiltonian that any two of the three zero-field split states are coupled by means of the magnetic field component pointing in the remaining third real-space dimension[32]. For example, $T_X$ and $T_Z$ are only coupled through $B_Y$, such that only the latter can drive the $T_X$–$T_Z$ transition. Because $x$, $y$ and $z$ are defined with

respect to the molecular axes, they might not coincide for different individual molecules (for example, see Fig. 4c).

## Origin of asymmetric lineshape

The hyperfine interaction in protonated pentacene can be described as an effective magnetic field $B_{HFI}$ created by the nuclei with non-zero spins acting on the electron spins. Assuming a random orientation of the 14 proton nuclear spins at a given point in time, $B_{HFI}$ will fluctuate around zero-field (see Extended Data Fig. 3c) and point in a random direction. Because of the many fluctuating nuclear spins acting together at random, the probability distribution of $B_{HFI}$ has its maximum around zero and falls off towards larger absolute values. The influence of $B_{HFI}$ is always small compared with the zero-field splitting such that $B_{HFI}$ shifts the energies of the triplet states only to the second order, that is, to $\propto B_{HFI}^2$ (ref. 21). This is depicted in Extended Data Fig. 3c for the $B_{HFI}$ component in the $z$ direction, $B_{HFI,z}$, in which it becomes clear that the broadening is single-sided in this case. The $x$ and $y$ components of $B_{HFI}$ contribute much less to the broadening. From Extended Data Fig. 3c, it becomes clear that the curvature around $B_{HFI} = 0$ of the hyperbolic avoided crossing is responsible for the asymmetric broadening. This curvature is inversely proportional to the energy difference of the respective pair of states. As the $T_X$–$T_Y$ transition has the smallest energy splitting of all possible pairs, the broadening is dominated by their avoided crossing occurring along the $z$ component of $B_{HFI}$ (ref. 21). Specifically, for the case of pentacene, this effect is smaller by roughly one order of magnitude in the other two directions.

Different individual isotopes contribute differently to $B_{HFI}$, such that different isotopologues give rise to a different probability distribution of $B_{HFI}$ when considering all possible nuclear spin configurations. The assignment to the isotopologue is done in comparison with previous work[29], based on the line profile, which not only includes the width but also its shape. Because the hyperfine interaction enters as a second-order term, the mere presence of one nucleus (for example, a proton) with strong hyperfine interaction also influences how strongly all the other nuclei (for example, deuterons) affect the line, thereby changing its overall shape.

Analogously, the hyperfine interaction of the eight proton nuclear spins in PTCDA gives rise to its asymmetric lineshape (see Fig. 2b). Note that the asymmetric shoulder appears here at the low-frequency side. Such a lineshape is expected for the $T_Y$–$T_Z$ signal[21], as becomes clear from Extended Data Fig. 3c (when considering the $T_Y$–$T_Z$ transition instead of the $T_X$–$T_Z$ transition that is explicitly illustrated). The $T_Y$–$T_Z$ signal is the largest for PTCDA; small signals were also observed for the $T_X$–$T_Y$ transition (at 252 MHz) and the $T_X$–$T_Z$ transition (at 1,501 MHz). Note that this is in contrast to pentacene, for which the $T_X$–$T_Z$ signal is the largest and the $T_Y$–$T_Z$ transition was not detected (because of the similarity in the lifetimes of its $T_Y$ and $T_Z$ states).

## Fitting of the lineshapes

As explained in the previous section, the hyperfine interaction is the origin of the asymmetric lineshape of the ESR signals. The lineshape of the $T_X$–$T_Z$ transition can be well approximated by a sudden onset at the frequency $f_{onset}$ followed by an exponential decay of width $f_{decay}$ (ref. 20) as

$$\Theta(f - f_{onset}) \exp(-(f - f_{onset})/f_{decay}),$$

in which $\Theta(x)$ denotes the Heaviside function.

A second contribution to the overall lineshape results from the finite lifetimes of the involved states. This leads to a lifetime broadening, resulting in a Lorentzian of the form

$$\pi^{-1} \Gamma / ((f - f_{res})^2 + \Gamma^2)$$

centred around each resonance frequency $f_{res}$ with a full width at half maximum of $2\Gamma$. Accordingly, the experimental resonances are fit

to a convolution of the above two functions, allowing to extract the broadening owing to the hyperfine interaction and the finite lifetimes separately. We note that non-Markovian processes[20,24] may lead to a deviation from the idealized Lorentzian and that power broadening was avoided in the measurements of the ESR-AFM signals. The effect of power broadening is illustrated in Extended Data Fig. 4.

## Rabi oscillations simulations and delay time

The Rabi oscillations were measured using an RF pulse applied around the middle of the dwell voltage pulse with a varying duration and a frequency corresponding to the maximum of the $T_X$–$T_Z$ or $T_X$–$T_Y$ ESR signal. To illustrate the effect of such an RF pulse for the $T_X$–$T_Z$ transition, the evolution of the populations of the three triplet states and the singlet state during the dwell voltage pulse were simulated, as shown in Extended Data Fig. 6.

These simulations were performed using the Maxwell–Bloch equations[40], analogous to the model used for ODMR[41]. The Rabi oscillation data are a temporal average of a single molecule, which—according to the ergodic assumption—is the same as an ensemble average. Therefore, we can use the density-matrix formalism[42] to simulate our data. Note that, on driving the $T_X$–$T_Z$ transition, $T_Y$ is decoupled from the $T_X$ and $T_Z$ dynamics and simply decays independently. The Bloch equations in the density-matrix formalism can therefore be restricted to the two coupled states[43], here $T_X$ and $T_Z$, whereas the occupation of the third triplet state is treated separately as a simple exponential decay function. With respect to the two coupled states, the system is described by the density matrix[42]

$$\rho = \begin{bmatrix} \rho_{ZZ} & \rho_{ZX} \\ \rho_{XZ} & \rho_{XX} \end{bmatrix}$$

and evolves according to the Liouville equation[42]

$$\frac{d\rho}{dt} = -\frac{i}{\hbar}[\mathcal{H}, \rho].$$

The Hamiltonian of the molecule interacting with the RF field (with Rabi rate $\Omega$) at resonance with the $T_X$–$T_Z$ transition (with resonance frequency $\omega_Z - \omega_X$) can be written as[41]

$$\mathcal{H} = \begin{bmatrix} \hbar\omega_X & -\hbar\Omega\cos(\omega_Z - \omega_X) \\ -\hbar\Omega\cos(\omega_Z - \omega_X) & \hbar\omega_Z \end{bmatrix}$$

The time evolution of the density operator in the rotating-frame approximation, with phenomenologically added relaxation and dephasing terms, can be described as[41]

$$\frac{d\rho_{XX}}{dt} = \frac{i\Omega}{2}(\rho_{ZX} - \rho_{XZ}) - \frac{\rho_{XX}}{\tau_X}$$

$$\frac{d\rho_{ZZ}}{dt} = \frac{i\Omega}{2}(\rho_{XZ} - \rho_{ZX}) - \frac{\rho_{ZZ}}{\tau_Z}$$

$$\frac{d\rho_{XZ}}{dt} = -\rho_{XZ}\left(\frac{1}{T_2} + \frac{1}{2\tau_X} + \frac{1}{2\tau_Z}\right) + \frac{i\Omega}{2}(\rho_{ZZ} - \rho_{XX})$$

$$\frac{d\rho_{ZX}}{dt} = -\rho_{ZX}\left(\frac{1}{T_2} + \frac{1}{2\tau_X} + \frac{1}{2\tau_Z}\right) + \frac{i\Omega}{2}(\rho_{XX} - \rho_{ZZ})$$

The time evolution of $T_Y$ is simply given by

$$\frac{d\rho_{YY}}{dt} = -\frac{\rho_{YY}}{\tau_Y}$$

These last five equations were used for the simulation for Extended Data Fig. 6. As input parameters for the simulation, we used the parameters that were experimentally derived for pentacene-$h_{14}$: the decay constants of the triplet states: $\tau_X = 20.8\ \mu s$, $\tau_Y = 66.6\ \mu s$ and $\tau_Z = 136.1\ \mu s$; the decay constant of the Rabi oscillations: $T_2 = 2.2\ \mu s$; the initial populations $\rho_{XX} = \rho_{YY} = \rho_{ZZ} = 0.8/3$ and coherences $\rho_{XZ} = \rho_{ZX} = 0$; the starting time of the RF pulse $t_S = 45.1\ \mu s$ and its duration of 4 and 4.5 Rabi-oscillation periods, respectively. Note that interconversion between $T_X$ and $T_Z$ resulting from spin-lattice relaxation is assumed to be negligible compared with $\tau_X$ and $\tau_Z$.

Here the initial occupation of the $T_X$, $T_Y$ and $T_Z$ states are assumed to be all equal to 0.8/3. Simulations and data in ref. 15 show that the triplet state is initially approximately 80% occupied (this value depends on the exact tip position, as the two competing tunnelling rates to form the $T_1$ and $S_0$ states depend on the wave-function overlap between tip and the LUMO and the HOMO, respectively). We assume that the probability to tunnel in the three states is equal (same spatial distribution and tunnelling barrier, as their energy differences are negligibly small). The Maxwell–Bloch simulations were performed to guide the understanding of our Rabi-oscillation measurements. For this purpose, we disregarded non-Markovian effects[20,24] and modelled the relaxation with a single phenomenological time constant $T_2$.

The delay time $t_S$, at which the RF pulses started, was fixed for one Rabi-oscillation sweep. The optimal $t_S$ was experimentally determined by sweeping the timing of a $\pi$ RF pulse over the range of the dwell pulse. A $t_S > 0$ is needed to initiate an imbalance between the $T_X$ and $T_Z$ states. Similarly, a decay time after the RF pulses is required such that the final triplet population is dominated by only one of these two triplet states. The optimal delay time is, therefore, shortly before the middle of the dwell voltage pulse. Furthermore, it is important that, on increasing the duration of the RF pulse, the sensitivity for differentiating $T_X$ and $T_Z$ does not greatly reduce, otherwise a further decay of the Rabi oscillations is induced by the readout. Therefore, we chose 30 $\mu s$ as a delay time for the Rabi oscillations of pentacene-$d_{14}$, which were probed up to an RF pulse duration of 30 $\mu s$.

## Rabi oscillations baseline fit

The baseline of the Rabi-oscillation experiment represents the situation of equal populations in the coupled states $T_X$ and $T_Z$ during the pulse; even if the Rabi signal is not yet decayed, it is oscillating around the baseline. The decay of the baseline arises from the decay of the (on average) equally populated $T_X$ and $T_Z$ states into the singlet state during the RF pulse. As the final population of $T_Y$ is independent of the RF signal, it will only give rise to a constant background and will be disregarded in the following.

Hence, the baseline is defined by the following: in the initial phase $0 < t < t_S$, all three triplet states decay independently from each other. At the beginning of the RF pulse, that is, at $t = t_S$, the sum of populations in $T_X$ and $T_Z$ is

$$P_{XZ}(t_S) = P_0/3 (\exp(-k_X t_S) + \exp(-k_Z t_S)),$$

in which $k_X = \tau_X^{-1}$ and $k_Z = \tau_Z^{-1}$ are the decay rates of $T_X$ and $T_Z$, respectively, and $P_0$ is the initial total population in the triplet state, such that $P_0/3$ is the initial population in each $T_X$, $T_Y$ and $T_Z$. During the RF pulse, that is, for $t_S < t < t_E$ (with $t_E$ being the end of the RF pulse), the RF signal equilibrates (on average) the populations of two of the states, thus at the end of the RF pulse

$$P_{XZ}(t_E) = P_{XZ}(t_S) \exp(-(k_X + k_Z)(t_E - t_S)/2).$$

Finally, for $t_E < t < t_D$, the states decay again independently, giving at the end of the dwell time

$$P_{XZ}(t_D) = P_{XZ}(t_S) \exp(-(k_X + k_Z)(t_E - t_S)/2)$$
$$\{\exp(-k_X(t_D - t_E)) + \exp(-k_Z(t_D - t_E))\}/2,$$

which can be rearranged to

$$P_{XZ}(t_D) = P_{XZ}(t_S)\{\exp(-k_X(t_D - t_S))\exp(t_{RF}(k_X - k_Z)/2)$$
$$+ \exp(-k_Z(t_D - t_S))\exp(-t_{RF}(k_X - k_Z)/2)\}/2.$$

Note that $P_{XZ}(t_S)$ does not depend on $t_{RF} = t_E - t_S$ and therefore just represents a constant prefactor. The two terms provide contributions to the baseline that rise and fall exponentially with $t_{RF}$, respectively. For the specific case and parameters considered here, the prefactor of the rising term is much smaller than that of the falling term and is therefore neglected. Because the decay rates were determined (Fig. 1b) for the pentacene-$h_{14}$ molecule, for which the Rabi oscillations were measured, these rates were used for the fitting of the Rabi oscillations of the pentacene-$h_{14}$ molecule (Fig. 3a). In case of pentacene-$d_{14}$ (Fig. 3c), we set $(k_X - k_Z)/2 = 0.012\ \mu s^{-1}$ based on the measured decay rates of another individual pentacene-$d_{14}$ molecule. In the experiment, other effects (for example, a thermal expansion owing to RF-induced heating) may also add to a temporal evolution of the baseline. These contributions were not separately accounted for but they are fitted as part of the falling term described above.

## Data availability

The data supporting the findings of this study are available from the University of Regensburg Publication Server at https://doi.org/10.5283/epub.54710 (ref. 44).

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

**Acknowledgements** We thank S. Fatayer, L. Gross, D. Peña, A. Donarini, F. Evers, J. Lupton and R. Huber for discussions and F. Bruckmann, T. Preis, C. Rohrer, C. Linz and D. Weiss for support. Funding from the ERC Synergy Grant MolDAM (no. 951519) and the Deutsche Forschungsgemeinschaft (DFG, German Research Foundation) through RE2669/6-2 is gratefully acknowledged.

**Author contributions** L.S. and J.R. conceived the experiment and L.S., R.S., S.B., J.E. and P.S. carried them out. L.S., R.S. and J.R. analysed the experimental results. L.S. and J.R. wrote the manuscript. All authors discussed the results and their interpretation and revised the manuscript.

**Competing interests** The authors declare no competing interests.

**Additional information**
**Correspondence and requests for materials** should be addressed to Lisanne Sellies or Jascha Repp.

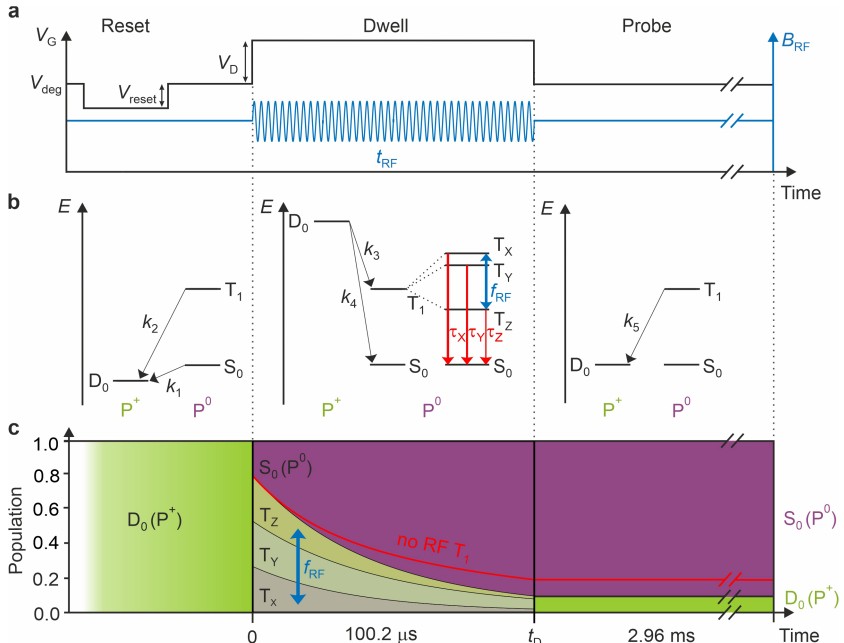

**Extended Data Fig. 1 | Schematic of the pump–probe sequence for the ESR-AFM measurements. a**, Voltage-pulse sequence (black) containing the reset pulse ($V_{reset}$ = −1.38 V, $t_{reset}$ = 33.4 μs) and dwell pulse ($V_D$ = 2.5 V, $t_D$ = 100.2 μs for pentacene, $V_D$ = 2.35 V, $t_D$ = 501 μs for PTCDA) applied as $V_G$ to the sample. During the probe interval (2.96 ms for pentacene), the voltage was set to the middle of the charging hysteresis of $S_0$ and $D_0$ ($V_{deg}$). An RF pulse (blue) was synchronized with the dwell pulse, having the same duration as the dwell pulse for the measurement of the ESR-AFM spectra. **b**, Many-body picture showing the mutual energetic alignment of the cationic (P$^+$) doublet ground state $D_0$, the neutral (P$^0$) singlet ground state $S_0$ and the neutral triplet excited state $T_1$ during the pump–probe sequence of **a**. The charge-transfer rates ($k_1$–$k_5$) were chosen to be much faster than the triplet decay, with $1/k_4$ set to around 4 μs

for the ESR-AFM experiments. During the reset pulse, the molecule was brought to $D_0$. The dwell voltage pulse lifted the $D_0$ state above the $T_1$ and $S_0$ states; an electron can tunnel into the molecule, forming either the $T_1$ or the $S_0$ state[15], preferentially occupying $T_1$ (see Methods), with rates $k_3$ and $k_4$, respectively. During the dwell pulse, the molecule in $T_1$ can decay into $S_0$. Two of the triplet states (here $T_X$ and $T_Z$) can be coupled during this time by the RF pulse. If the molecule was still in the triplet state after the dwell pulse, an electron can tunnel out of the molecule charging it, allowing a discrimination of the triplet and singlet states through the charged and neutral states, respectively. **c**, Populations of the involved states during the pump–probe sequence, with and without RF. Note that, with RF, the $T_X$ and $T_Z$ states decay with the average decay rates of $T_X$ and $T_Z$, assuming a sufficiently strong RF pulse.

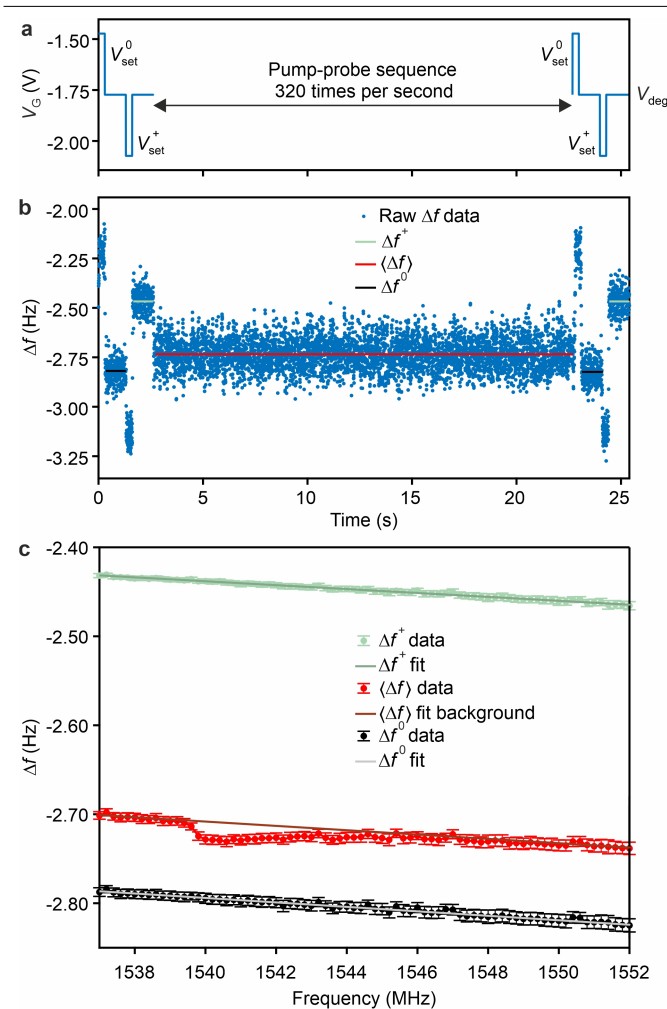

**Extended Data Fig. 2 | Raw data and signal extraction towards an ESR spectrum. a**, Voltage-pulse sequence for the acquisition of one data point. At the beginning and end, two voltage pulses were given to neutralize ($V_{set}^0 = V_{deg} + 0.3$ V) and charge ($V_{set}^+ = V_{deg} - 0.3$ V) the molecule, in between which the voltage was set to the centre of the charging hysteresis ($V_{deg}$); here a typical value for pentacene is shown. During the middle 20 s of the data trace, the pump–probe sequence shown in Extended Data Fig. 1a was repeated 320 times per second. **b**, One of the recorded $\Delta f$ data traces with the pulse sequence shown in **a**. The frequency shifts of the neutral ($\Delta f^0$, black) and charged ($\Delta f^+$, green) molecule were extracted as the average over the 1-s intervals at the beginning and end of the trace. The averaged frequency shift ($\langle \Delta f \rangle$, red) was extracted from the interval during which the pump–probe sequence was turned on. **c**, ESR spectrum of pentacene-$h_{14}$ without normalizing the frequency shift. The panel shows $\Delta f^0$, $\Delta f^+$ and $\langle \Delta f \rangle$ as a function of the RF (error bars are s.d. of seven repetitions). Owing to slight creep and drift, all three signals show a similar overall trend line (fit curves). Around 1,540 MHz, $\langle \Delta f \rangle$ shows clear deviations from this general trend, representing the ESR signal. Normalizing the frequency shift from these three values as $\Delta f_{norm} = \frac{\langle \Delta f \rangle - \Delta f^0}{\Delta f^+ - \Delta f^0}$ largely reduces the background trend owing to creep and drift.

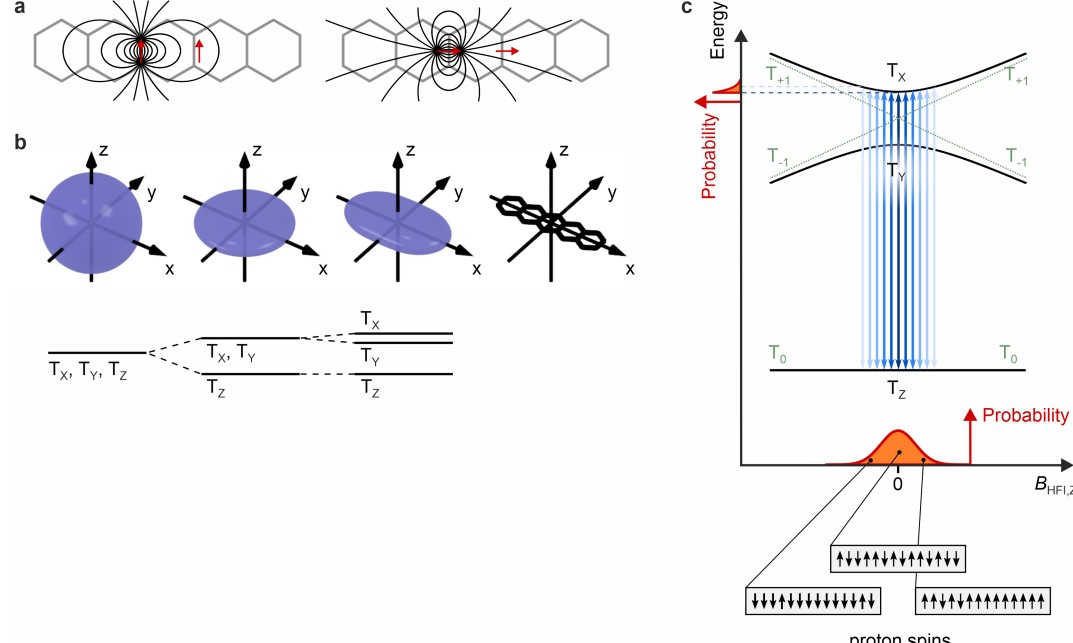

**Extended Data Fig. 3 | Illustration of the zero-field splitting and explanation of the asymmetric lineshape. a**, Schematic illustration of the anisotropic nature of magnetic dipole–dipole interaction (black field lines) between the two spins (red arrows) constituting the triplet state, as shown for the case of pentacene (grey molecular skeleton). **b**, For a spherical density distribution of the two electrons, the three triplet states $T_X$, $T_Y$ and $T_Z$ are degenerate. However, for an oblate density, the probability distribution of the electrons' mutual distance differs in the different spatial directions. In this case, because of the anisotropy of the dipole–dipole interaction, the alignment of the spins with respect to the spatial directions matters and $T_Z$ splits off in energy. For a probability distribution that differs in all three dimensions, the degeneracy of all three states ($T_X$, $T_Y$ and $T_Z$) is lifted. In these considerations, the $x$, $y$ and $z$ directions refer to the high-symmetry directions of the molecule, as depicted[21]. **c**, The hyperfine interaction in protonated pentacene can be described as an effective magnetic field $B_{HFI}$ created by the nuclei acting on the electron spins. Owing to the random orientation of the 14 proton nuclear spins (bottom), $B_{HFI}$ will fluctuate around zero-field and its probability distribution for the $z$ component is depicted (orange, bottom). The $T_X$ and $T_Y$ states show a hyperbolic energy dependence (top) as a function of a magnetic field in the $z$ direction, $B_{HFI,Z}$. Weighting the different transition frequencies (blue double-headed arrows) with the probability distribution of $B_{HFI,Z}$ gives rise to the asymmetric lineshape as schematically illustrated for the $T_X$–$T_Z$ transition by the projection onto the energy axis (orange, top left). Of all directions, the $z$ component is the most important one for the hyperfine-related lineshape (see Methods).

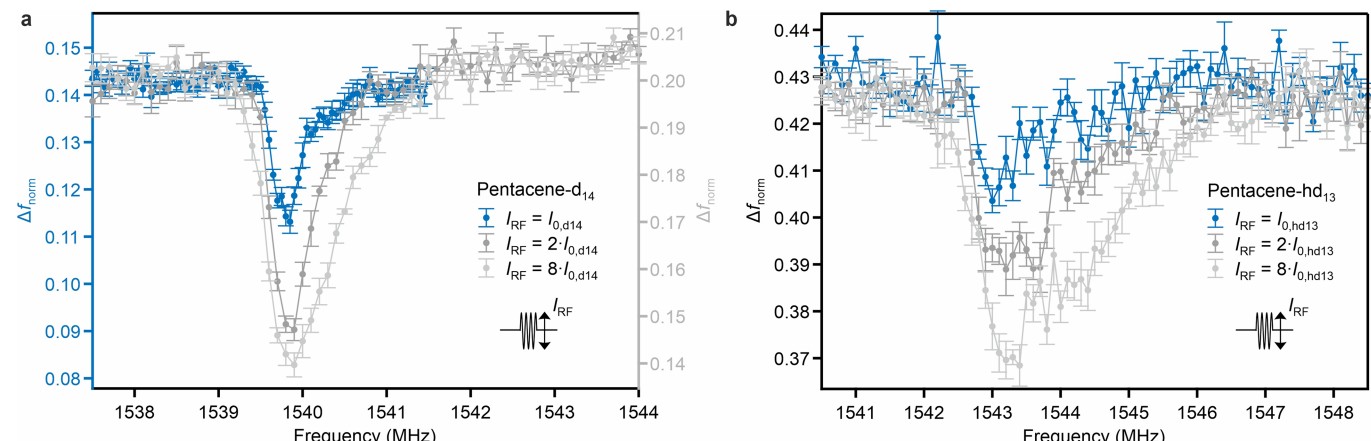

**Extended Data Fig. 4 | Power broadening of ESR spectra. a,b**, Spectra acquired at different RF powers demonstrating the role of power broadening for pentacene-$d_{14}$ and pentacene-h-$d_{13}$, respectively. For the spectra at medium and large powers, the amplitude of the RF pulse was doubled and octuplicated, respectively. Although from low to medium power the ESR signal is mainly rescaled in amplitude, at the largest power, the broadening becomes recognizable. When comparing different lineshapes, the exclusion of power broadening is necessary[20,29]. The error bars were derived from the s.d. of 12 repetitions for the lowest power of pentacene-$d_{14}$ and six repetitions for the other datasets.

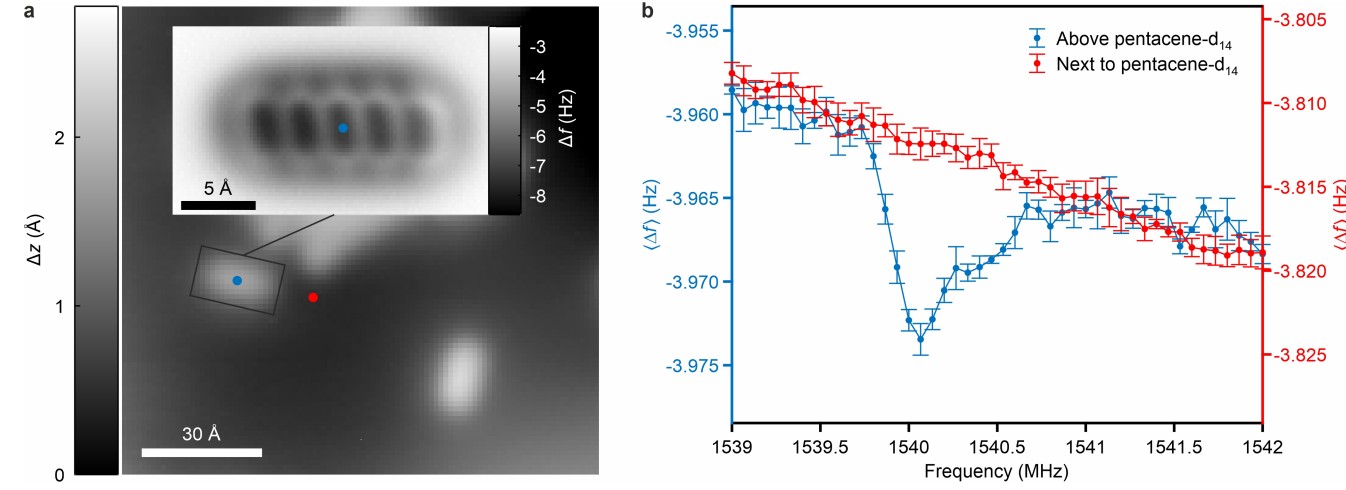

**Extended Data Fig. 5 | Spectra above and next to a pentacene molecule.**
**a**, AFM topography image of a surface area with adsorbed pentacene molecules measured with a CO-terminated tip (set point: $\Delta f$ = −1.45 Hz at $V$ = 0 V, $A$ = 1.65 Å). The inset shows a constant-height AFM image as a zoom-in ($A$ = 0.3 Å, $\Delta z$ = −4.76 Å with respect to the set point $\Delta f$ = −1.45 Hz at $V$ = 0 V, $A$ = 0.3 Å). **b**, ESR-AFM spectra above (blue) and next to (red) the pentacene-$d_{14}$ molecule shown in **a** (error bars are s.d. of four repetitions). Although the former shows a clear feature around 1,540 MHz, this feature is absent for the spectrum acquired next to the molecule. For a discussion of the spatial confinement of the ESR-AFM signal, see the main text. Note that the frequency shift was not normalized for these spectra.

**a**

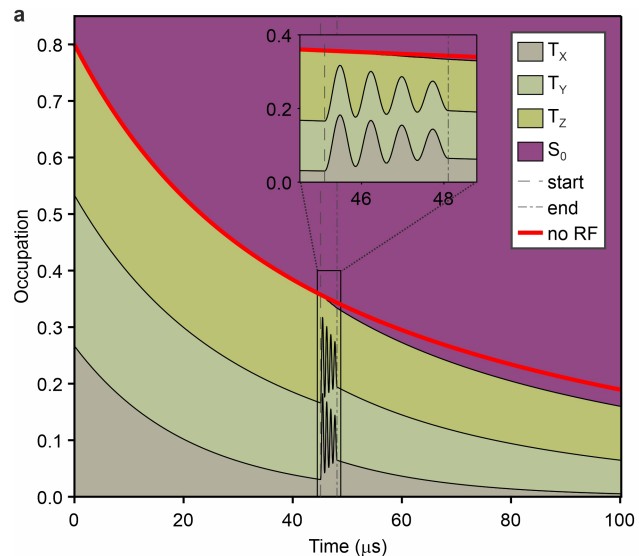

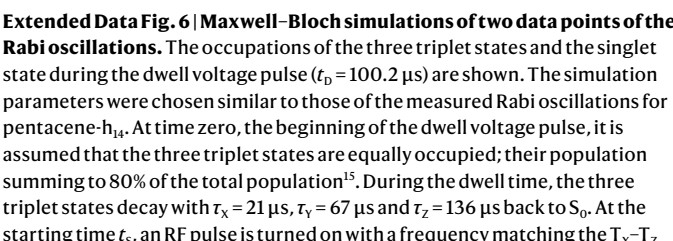

**b**

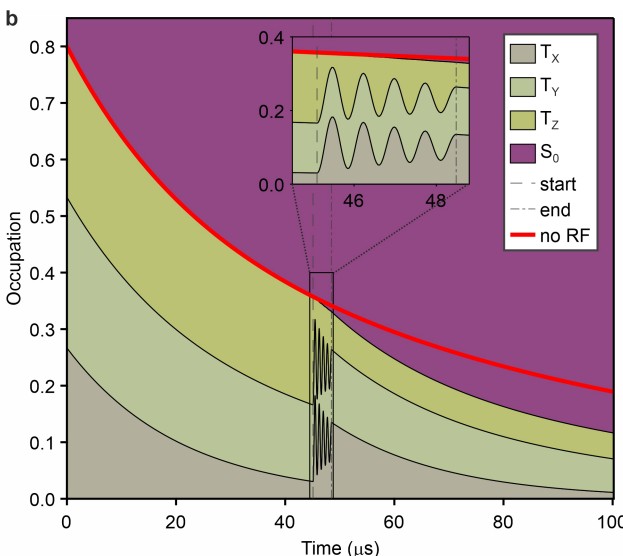

**Extended Data Fig. 6 | Maxwell–Bloch simulations of two data points of the Rabi oscillations.** The occupations of the three triplet states and the singlet state during the dwell voltage pulse ($t_D = 100.2$ μs) are shown. The simulation parameters were chosen similar to those of the measured Rabi oscillations for pentacene-$h_{14}$. At time zero, the beginning of the dwell voltage pulse, it is assumed that the three triplet states are equally occupied; their population summing to 80% of the total population[15]. During the dwell time, the three triplet states decay with $\tau_X = 21$ μs, $\tau_Y = 67$ μs and $\tau_Z = 136$ μs back to $S_0$. At the starting time $t_S$, an RF pulse is turned on with a frequency matching the $T_X$–$T_Z$ energy splitting. This RF pulse causes coherent oscillations between these two triplet states, as clearly visible in the inset. The population in the $T_X$ and $T_Z$ states at the end of the RF pulse depend, thus, on the duration of the RF pulse. The larger the population in the fastest-decaying $T_X$ state, the lower the triplet population at the end of the dwell pulse. This is exemplified by the simulations shown in **a** and **b** with RF pulse durations corresponding to 4 and 4.5 Rabi-oscillation periods, respectively (Rabi frequency: 1.33 MHz, decay constant of the oscillations: 2.2 μs).

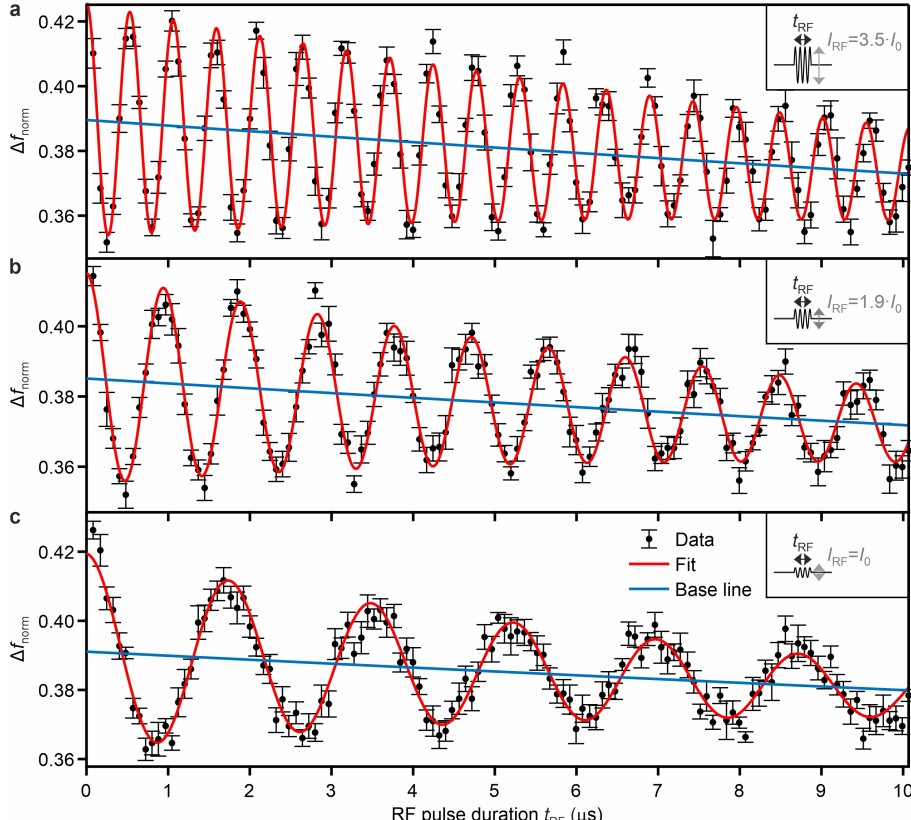

**Extended Data Fig. 7 | Power dependence of Rabi oscillations. a–c**, Rabi oscillations acquired on pentacene-$d_{14}$ with a CO-terminated tip at different RF amplitudes demonstrating the increase of Rabi frequency with increasing RF amplitude (error bars are s.d. of eight repetitions). The Rabi frequency increases (almost) linearly with amplitude, which is characteristic for Rabi oscillations. The slight deviation from a linear increase is attributed to nonlinearities of the RF circuitry.

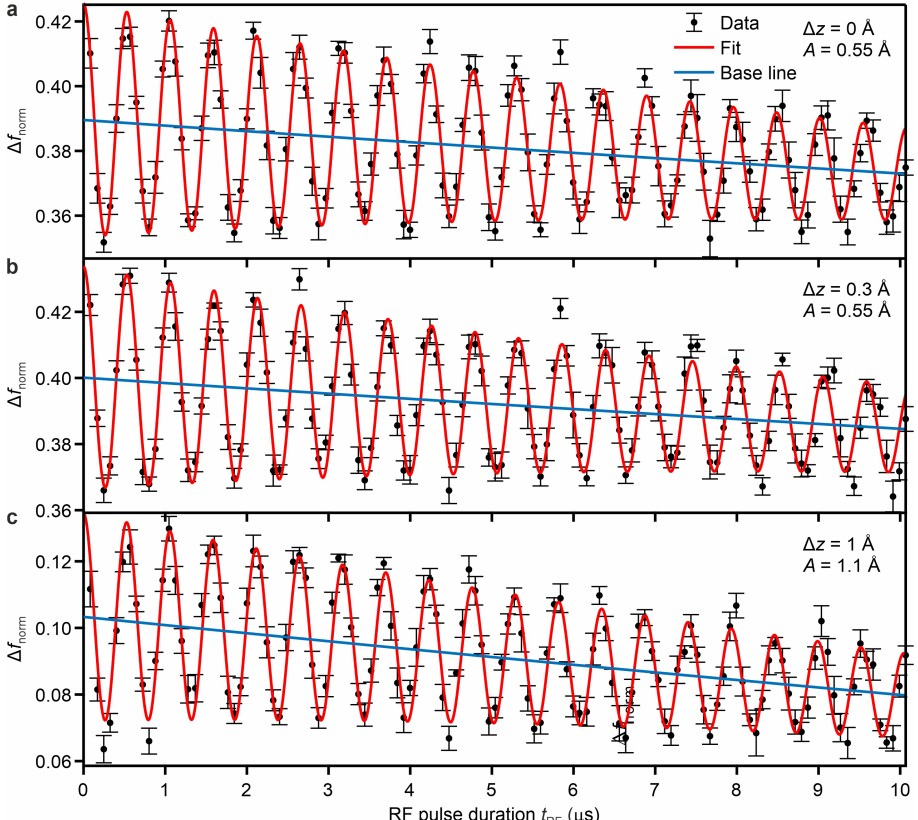

**Extended Data Fig. 8 | Role of tip height and cantilever amplitude on Rabi oscillations. a–c**, Rabi oscillations acquired on pentacene-d$_{14}$ with a CO-terminated tip at relative different tip heights $\Delta z$ (referring to the zero crossing from the set point of the cantilever: $\Delta f = -1.42$ Hz at $V = 0$ V, $A = 1.65$ Å; positive $\Delta z$ values are further away from the surface) and cantilever oscillation amplitudes $A$, as indicated in the panels (error bars are s.d. of eight repetitions). The Rabi oscillations show no appreciable differences.

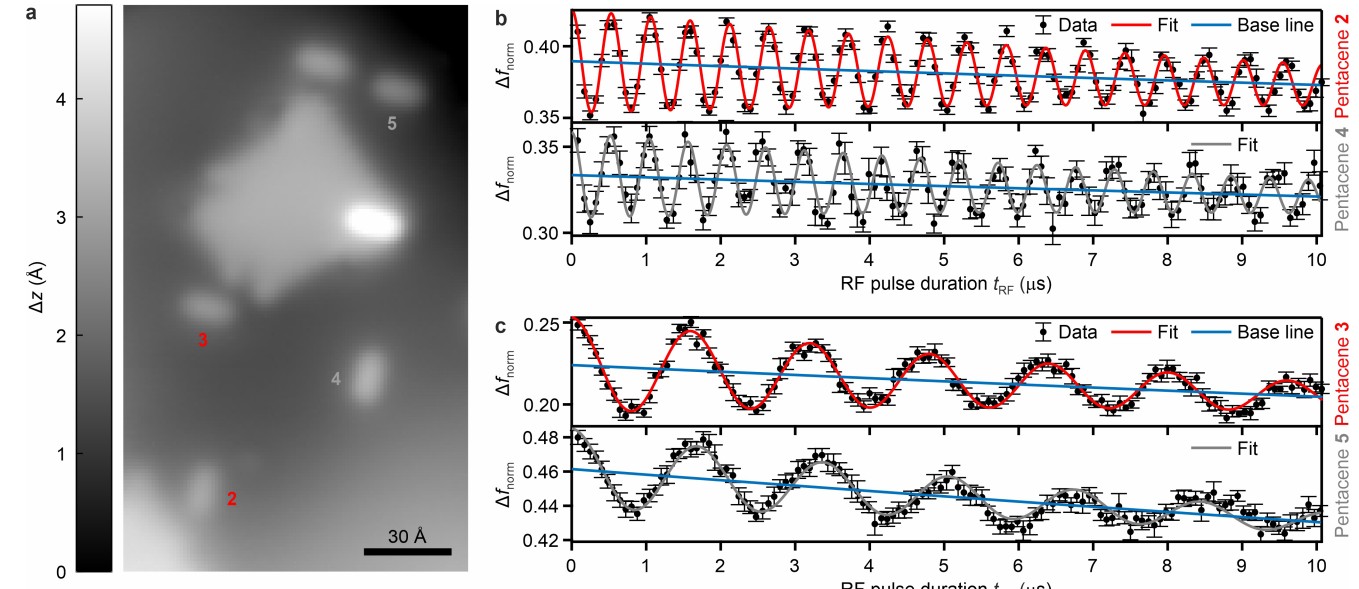

**Extended Data Fig. 9 | Demonstration of orientational dependence of Rabi oscillations for a set of molecules. a**, AFM topography image of a surface area with adsorbed pentacene molecules measured with a CO-terminated tip (set point: $\Delta f = -1.45$ Hz at $V = 0$ V, $A = 1.65$ Å). **b,c**, Rabi oscillations of the $T_x$–$T_z$ transition of four pentacene-$d_{14}$ molecules, two per orientation, exemplarily to demonstrate the reproducibility of the determined Rabi frequencies for the two orientations. The averaged values of the Rabi frequencies are $1.90 \pm 0.02$ MHz and $0.60 \pm 0.02$ MHz, derived for four and three molecules, respectively. The error bars on the data points were derived from the s.d. of eight repetitions.

**Extended Data Table 1 | Molecule-to-molecule variations of the resonance frequencies**

| Tip | Pentacene-$d_{14}$ | $T_X$-$T_Y$ Transition | $T_X$-$T_Z$ Transition |
|---|---|---|---|
| a (metal tip) | 1 | 118.1 MHz | 1538.2 MHz |
| | 2 | 118.2 MHz | 1538.5 MHz |
| | 3 | 118.0 MHz | 1539.1 MHz |
| | 4 | 117.4 MHz | 1539.9 MHz |
| | 5 | 116.4 MHz | 1540.2 MHz |
| b (CO tip) | 6 | 116.9 MHz | 1540.1 MHz |
| | 7 | 116.8 MHz | 1540.0 MHz |
| | 8 | 118.6 MHz | 1539.6 MHz |
| | 9 | 117.1 MHz | 1540.2 MHz |
| | 10 | 117.0 MHz | 1539.9 MHz |
| | 11 | 116.6 MHz | 1540.3 MHz |

List of the $T_X$–$T_Y$ and $T_X$–$T_Z$ transition frequencies for 11 different individual molecules, showing small but appreciable molecule-to-molecule variations. We attribute these small differences to variations of the local environment. For example, the data hint to correlations between the observed differences in resonance frequencies and the existence of nearby step edges of NaCl, which is subject to further investigations. For data acquisition, two different tips were used as indicated, of which one was terminated with a CO molecule. Note that the atomic structure of the tip apex was not altered between the different datasets corresponding to the same tip.