## [Peer Review File · Nature]

Manuscript Title: Single-molecule electron spin resonance by means of atomic force microscopy

Reviewer Comments & Author Rebuttals

Reviewer Reports on the Initial Version:

Referee #1 (Remarks to the Author):

1 In the manuscript from Sellies, et al, the authors report on the coherence time of an individual pentacene molecule on the surface of insulating NaCl based on a newly developed ESR-AFM technique. In a previous report (ref 22), the group reported on singlet-triplet excitations of the same pentacene type molecule using a charge transfer technique on a thick NaCl substrate. Based on the observation of long lifetimes of the excited triplet states, in this work they adapt a strip line next to the sample and apply continuous wave RF excitation as a function of frequency. By measuring the frequency shift of the oscillating AFM tip, they see resonance like features in the frequency shift at specific frequencies that correlate with the expected energy differences of the excited triplet state. This is done in the absence of magnetic field, but is not essential due to the zero-field splitting of the triplet state. Based on this excitation, they perform particular pulsed experiments to measure Rabi oscillations and extract a coherence time of the molecule, which is on the order of microseconds. They also measure compare the measured resonance linewidth of the same molecule with a different isotope, where they observe a clear difference in the resonance behavior. Based on this, they argue that the nuclear moments of the molecule play a role in decoherence and also extract a coherence time based on the aforementioned Rabi experiment. Overall assessment and recommendation: The experimental data in this paper is eye popping and very provocative, pushing the state of the art to a new realm. The data is extremely beautiful and impressive, and some of the best experimental data I have seen in the last few years. However, I cannot recommend this for publication in Nature, and suggest this be transferred to a journal like Nature Physics or Nature Nanotechnology, pending much needed changes and expanded discussion. The work is sufficiently differentiated from previous works' involving ESR-STM, and has extremely high impact potential. But I do not feel the results shown have realized such potential to warrant publication in Nature. In the end, the biggest piece missing is any sort of spatial resolution, e.g. intramolecular, to call this microscopy. I firmly believe spatial resolution with this method is a game changer, especially if one can learn about nuclear moments, structure, and coherent dynamics complementary to what ncAFM can do. I also want to clearly add: I found the paper far too short and lacking many relevant details. The paper in its current state has a bit of a mixed identity. It sits in between being a paper a) about a new and game changing method, b) and being a paper about a long-lived molecular qubit. But the paper is not really either. Concerning a) it lacks a lot of relevant details and benchmarking; the former may limit other groups to reproduce the method (detailed below). It also compares itself to other methods, but I was skeptical in the way this was done (apples to oranges). On the other hand, b) the paper is focused on long coherence times rather than what more can be learned from the spectra about the molecule. Unfortunately, this molecule has already been measured 30 years ago, as cited. And I tried very hard and eagerly to constructively find what new insight this method brings to the table, when compared to the 1993 Nature paper they cite. But I could not really find any. I'm

not sure the scientific find that a single molecule on a thick insulator has long coherence times is a Nature worthy claim. I also do not think SPM is going to be the basis of a new generation of coupled qubit technology. But that is a bit of the impression of the tone I get from this paper (especially the intro and the focus in the figures). I may be wrong, but I don't think the qubit community will be impacted by these findings. Coupled qubits have been made from atoms (e.g. dopants in semiconductors, Rydberg atoms) and molecules (e.g. molecular magnets). I believe the impact of this work is in the method, and what can be learned from it. The paper would benefit from a focus on this, and its perspectives in juxtaposition to other quantum sensing methods. Below, I also include detailed scientific comments and questions that should be addressed in any future revision.

Detailed points/questions:

- Do the ESR spectra depend on where the probe is placed, with respect to the molecule or distance from the molecule?
- What does the imaging at the apparent measurement conditions look like?
- I had a hard time understanding what is actually measured, i.e. what is df_{norm} . Are the curves in e.g. Fig. 2a, what is directly measured in the df signal? If this is directly measured, then why would the force change in resonance? Or is this indirectly measured via this charge state, and thus extracted?
- 115MHz detection corresponds nearly to an equilibrium spin temperature of mK. Why can this be detected nearly three orders of magnitude higher in temperature? For example, I believe all triplet states are populated after excitation. But, isn't there a thermal probability to repopulate all triplet states after one of them decays?
- The authors do not provide any imaging and details about the NaCl growth. Are there defects, is it polycrystalline? Will the quality of the NaCl (or lack of it), lead to variations between measured molecules (e.g. due to defects)?
- Statement about Fig. 2A being related to hyperfine is not clear to me: does this lineshape depend on applied power/amplitude? Can this be better explained (or backed up by some experimental evidence)?
- The way the striplines are designed, are there field gradients and certain selection rules to be concerned with? What is the actual direction of the AC field with respect to the molecule? Does the molecular orientation matter?
- How is the transmission of the stripline calibrated?
- What determines the error bars in the df signal in all this data? Are these error bars only for a single measurement, and all are otherwise the same, i.e. it's extremely reproducible from molecule to molecule?
- How many tips/molecules were measured, are they all the same, or are there variations or tips that doesn't show resonance? Does every probe show a resonance? How are the probes created? These are important details when benchmarking a new method.
- "This splitting arises predominantly from the dipole-dipole interaction of the two unpaired electron spins. It is therefore governed by the spatial distribution of the triplet state and can serve as a fingerprint of the molecule."
- Likewise, won't this lead to a distortion of the molecule what the AFM may not see but can be learned about from the spectral data? Also when it is charged?
- What are the crystal field and spin Hamiltonians?
- What creates the zero-field splitting and why is there a difference between X, Y?
- "The shift of the TX-TZ transition frequency by 60 MHz (~ 4% of its value) with respect to pentacene molecules in a host matrix^{24,25} can be rationalized by the different environments. In fact, the adsorption on a surface is expected to affect the triplet state electronic distribution along the z-direction (perpendicular to the surface), and thereby the energy of TZ."
- This was confusing to me. What give rise to the variations? What is the role of the ionic environment of NaCl, and may this play a role, similar to what is claimed in ref. 9?
- Can the two states be indicated in Fig. 2c? Not clear what the two states on the Bloch sphere are here, as there are multiple excitations. A picture and the relevant states would be useful.
- I find this statement a bit unscientific in the course of reading the paper: "To demonstrate the enhanced coherence that results from our minimally perturbing detection scheme, we measured Rabi oscillations." 1) I don't

know what enhanced coherence means. 2) To make this statement, a) the decoherence mechanism needs to be known, b) the “unperturbed” coherence time of that state needs to be known, and then shown that this method does not affect it. • The simulations in Extended Fig. 3 are quite confusing. How is this calculated? Also, how are the occupations known, what Hamiltonian is expected and how is the effect of pumping considered in the problem? Is thermal broadening considered in this calculation? • I also do not fully understand what is shown in Extended Fig 4? • “Similarly, the Rabi oscillations represent a time average of fluctuating nuclear spin configurations and, consequently, of a fluctuating Rabi frequency. Hence, the observed decay of the oscillations 3 is likely dominated by the dephasing from the fluctuating resonance frequency, limiting the coherence time²⁹. Reducing the hyperfine interaction should then increase the coherence time further.” o Is this contradictory, or is it the statement that exchange with the hyperfine moments create dephasing? The way it reads now, it sounds like the resonance frequency is not cleanly applied. • “The left flank of the signal corresponds to a broadening of 0.13 MHz (full width at half maximum, see Methods), being strongly reduced compared to ESR-STM⁷” o I found the comparison to ESR-STM apples to oranges. I think a comparison to ESRSTM can only be made if both methods are applied on one and same sample system. As the mechanisms are quite different, I do not find it appropriate to make direct comparisons which are better/worse. Likewise, if there is no spatial resolution here, this is clearly an advantage of a method like ESR-STM. • “This sharp onset corresponds to a sub-nanoelectronvolt resolution in energy, enabled by eliminating main decoherence sources.” o I do not think one can claim nanoelectron volt resolution in a non-equilibrium experiment, as this may violate thermodynamic arguments, especially asking the question about what temperature means in this system. • Does the Q factor of the oscillator change during the measurement, or is there any measurable dissipation/excitation signal? • At the end of page 5 and top of page 6, the authors point out decoherence mechanisms of other methods, but nothing about their own. Unless the authors want to claim that this is a perfect unperturbative method, then there also needs to be some text describing potential sources of decoherence from the measurement method, as this is the first paper. o For example, if the tip itself has an electric dipole moment, it certainly can lead to dephasing of the molecule and potentially decoherence, right? Or are all probes prepared in the same way and have no effect? o Likewise, the authors should also check other potential sources: Brownian noise of an oscillator for example (or amplitude/tip termination dependence). I understand what the authors want to claim here, but this is too strong of a statement. o In the same way, I think this sentence could be a bit softer as it’s a suggestion: The long coherence time observed here is close to this intrinsic limit. This strongly suggests that decoherence due to the currentless but force-detected ESR, as introduced here, is negligible. Thus, ESR-AFM may even allow for coherent electron spin manipulation and read out at millisecond time scales^{24,32}, while providing atomic-scale real-space control^{22,33,34}. • “Further, the thick NaCl film completely decouples the molecule from the conducting substrate, eliminating decoherence caused by scattering electrons from the substrate^{11,18,31}” • Again, I find this statement too strong. Indeed, there’s no electron transport, but the molecule is not completely decoupled from the film, as it is a dielectric environment. The authors already site an example of where the results differ from previous pentacene measurements. Also, the underlying film is ionic. • Much of this paper assumes the reader has digested the group’s previous Science paper. The text would benefit a lot from some better illustration and clearer descriptions. Can these charge states be labeled in Fig. 1, and where the unpaired spins are? • In 1b, what does a broadband pulse mean, isn’t this continuous wave excitation? • “This reduction is due to the smaller hyperfine interaction of deuterium; the width is now dominated by nuclear electric

quadrupole interaction." This is an example of a suggestive statement where there is not direct experimental evidence: the authors do not measure this hyperfine interaction. I would tone down such statements, e.g. this reduction may be due... • Refs. 15-16 in the abstract are misleadingly cited, none of this work is related to a quantum device. 4

Referee #2 (Remarks to the Author):

In "Atomic-scale access to long spin coherence in single molecules with scanning force microscopy" (2022), Sellies, Spachtholz, Scheuerer & Repp demonstrate single-molecule spin resonance that is detected for the first time in a low-temperature AFM setup, where decoherence due to tunnel current or a magnetic tip are essentially absent. This work is related to several previous lines of work, which are well referenced: the detection of spin resonance among the triplet states in individual pentacene molecules, which was optically detected since the 1990's; recent work by the Repp group observing spin relaxation from the same triplet states of the same molecule using AFM; use of AFM to detect single-electron spin resonance in magnetic resonance force microscopy (MRFM) in the early 2000's; and electrically-driven spin resonance in STM observed by several groups. This paper achieves the notable experimental feat of measuring clear Rabi oscillations among the triplet spin states, with coherence times exceeding those from recent STM-based work by more than a factor of ten, in a setup that can potentially give local atomic resolution as seen in this group's previous AFM work on pentacene on thick NaCl. On a technical level, this experimental effort combines an innovative initialization and detection schemes that combine tip gating with spin-to-charge with charge-to-force conversion. It makes use of careful and elaborate set of techniques for pulse timing, synchronization, noise cancellation, and drift compensation to obtain clear and convincing signals.

It is an impressive surprise that it is possible to observe single-molecule spin resonance in this force-detection scheme, where no net tunnel current, magnetic tip, single-electron transistor, or optical signals are used. In this way, this work is groundbreaking. However, it is not clear how widely applicable the technique may be, and will be quite challenging for non-specialists to follow the main physics and techniques involved, in order to imagine its use outside this specialized system. Since it does not appear to show important new properties of the molecular spin studied, this is primarily a paper about the new technique, and would need to characterize the technique in more depth and show that it applies to a variety of other systems of possible interest. My assessment is this paper surely does belong in a high-quality journal but does not meet the very high level of broad accessibility or impactful applicability that is required for a Nature publication. In any case, the following comments and questions may help to improve the manuscript.

It takes quite a bit of effort in studying this paper to feel confidence that the observed Rabi signals cannot be due to some other oscillatory property, such as an electronic oscillation, mechanical mode of the cantilever, or of surface acoustic waves in the Au microstrip layer, as seen for example in the MHz surface acoustic waves in gold films by [Sheng et al. PRL 129, 043001 (2022)]. After examining the data that is shown, I am quite convinced that ESR and Rabi oscillation is observed, but it would be more immediately persuasive if it were shown that the Rabi frequency is proportional to the RF amplitude. I encourage the authors to include that measurement if it is available. Such a measurement would also serve as a basic control and characterization test that many readers would expect to see for this scheme.

Please give an idea of how broad the data set is. For how many molecules was spin resonance observed for each isotopologue?

The reduction in line width when using deuterated molecules is compelling, and is compatible with previous ensemble ESR. However, it is confusing why the h-d13 line here seems to be essentially

the same width (~ 3 MHz) as for h14. How is this possible? How is it determined that the molecule is an instance of h-d13 rather than an errant h14 found due to contamination from previous doses? Is there any measurement besides the line width that supports assignment of the isotopologues?

It is not clear how broadly applicable the technique can be. The triplet excited state in which splittings are set by the molecular anisotropy rather than by Zeeman energies. The pentacene triplet states are a well investigated and historically important model spin system for previous single-molecule ESR, but it would seem to be a fairly special case. How broad a range of molecular spins have accessible triplet states, long enough state lifetimes, sufficiently distinct relaxation times, and nearby states for spin-to-charge conversion, all of which are required here? Alternatively, is there any clear path to adapting this method to other kinds of molecular or atomic spins?

The usual spin-1/2 ESR is sensitive directly to B fields and couplings in well established ways, but the pentacene transitions are insensitive to B fields to first order, giving them good coherence but making them poor as sensors. On the other hand, as qubits for quantum information they would need to be coupled together and sensitive to their neighbors. Please mention how these could plausibly be applied as detectors, or as long-coherence coupled qubits, or in some other way.

A description of the quantum states T_x , T_y , and T_z is needed, along with the Hamiltonian leading to them and the resulting selection rules for coherently driving transitions among them given the in-plane RF magnetic field. What model was used for the Rabi oscillation part of the Maxwell-Bloch simulations? Can the orientation of the molecules be deduced from the Rabi rates and does this agree with the imaged orientation? Reference [25] (Kohler et al 1993) section 2.2 appears to be a good presentation of the states, and the authors of this paper could refer readers directly there, or even better, spell it out in their own terms. If I understand correctly from [25], the triplet states can be written (unnormalized, and using the surface-normal z basis) as $T_x = 11 - 00$; $T_y = 11 + 00$; and $T_z = 10 + 01$. The driving field BRF would need to create an oscillating energy difference between the sums and differences of the two states involved in a transition. For example, driving between T_x and T_y happens when the state $T_x + T_y = 11$ differs in energy from $T_x - T_y = 00$, and this indeed is the case for a B field in the z direction. However, the strip line appears to give an essentially xy -plane B field, so please explain how this transition is driven coherently. A similar analysis for the T_x to T_z transition is also needed.

The highly-asymmetric line shape for h14 pentacene is well established from previous publications, but for general readership it would help to provide a qualitative idea why the hyperfine coupling gives such asymmetry.

In the only AFM image (inset greyscale image of Fig. 3c) of the topography, it is not very convincing that individual molecules are observed. It is helpful that previous work by this group on triplet quenching showed that they are capable of discerning the molecular orientation and atomic environment using AFM alone. It would strengthen this paper to show clearer images, and showing that the presence or absence of the ESR and Rabi signals correlates with presence of the discernable molecule and its orientation.

In Ext. Data Fig. 2, should $\langle \Delta f_0 \rangle$ be simply $\langle \Delta f \rangle$?

The word "broadband" in Fig.1 caption is confusing. Perhaps it means "radio-frequency", at a single frequency f_{XZ} .

Author Rebuttals to Initial Comments:

Referee #1

R1: The experimental data in this paper is eye popping and very provocative, pushing the state of the art to a new realm. The data is extremely beautiful and impressive, and some of the best experimental data I have seen in the last few years.

Authors: *We thank the referee for the appreciation of our work and these very positive statements about it. We also thank the referee for the time and effort invested into the review and for the detailed comments.*

R1: However, I cannot recommend this for publication in Nature, and suggest this be transferred to a journal like Nature Physics or Nature Nanotechnology, pending much needed changes and expanded discussion. The work is sufficiently differentiated from previous works' involving ESR-STM, and has extremely high impact *potential*. But I do not feel the results shown have realized such potential to warrant publication in Nature. In the end, the biggest piece missing is any sort of spatial resolution, e.g. intramolecular, to call this microscopy. I firmly believe spatial resolution with this method is a game changer, especially if one can learn about nuclear moments, structure, and coherent dynamics complementary to what ncAFM can do. I also want to clearly add: I found the paper far too short and lacking many relevant details.

The paper in its current state has a bit of a mixed identity. It sits in between being a paper a) about a new and game changing method, b) and being a paper about a long-lived molecular qubit. But the paper is not really either. Concerning a) it lacks a lot of relevant details and benchmarking; the former may limit other groups to reproduce the method (detailed below). It also compares itself to other methods, but I was skeptical in the way this was done (apples to oranges). On the other hand, b) the paper is focused on long coherence times rather than what more can be learned from the spectra about the molecule. Unfortunately, this molecule has already been measured 30 years ago, as cited. And I tried very hard and eagerly to constructively find what new insight this method brings to the table, when compared to the 1993 Nature paper they cite. But I could not really find any. I'm not sure the scientific find that a single molecule on a thick insulator has long coherence times is a Nature worthy claim. I also do not think SPM is going to be the basis of a new generation of coupled qubit technology. But that is a bit of the impression of the tone I get from this paper (especially the intro and the focus in the figures). I may be wrong, but I don't think the qubit community will be impacted by these findings. Coupled qubits have been made from atoms (e.g. dopants in semiconductors, Rydberg atoms) and molecules (e.g. molecular magnets). I believe the impact of this work is in the method, and what can be learned from it. The paper would benefit from a focus on this, and its perspectives in juxtaposition to other quantum sensing methods.

Authors: *We are grateful that the basis of the referee's overall assessment is explained, giving us the opportunity to provide our perspective on the matter. Indeed, we view our work being of the referee's category a: "a new and game changing method." At the same time, we are convinced that a high-impact publication introducing a novel method should always demonstrate the method's strength while addressing a relevant system/application. We neither wanted to claim that our method will be used in the near future to build actual large-scale quantum computers, nor did we want to claim that the observed coherence time alone justifies a Nature publication. We clarified this in the revised version. The referee rightfully asks what new insight this method brings to the table, when compared to the 1993 Nature papers:*

It is the possibility to image and characterize the unique quantum system under consideration in all its atomic details. The referee is certainly aware, that every two individual solid-state-based quantum-dots behave different, even if they are nominally identical. This poses a big challenge for the fundamental understanding of the different sources of quantum decoherence and even more so for device integration and upscaling.

Being able to specify the atomic environment of a given individual quantum dot and at the same time characterize its quantum properties will for the first time provide a direct link how the environment leads to decoherence. Combined with atomic manipulation techniques, this may eventually even allow to modify minute details in the atomic environment and to directly observe its impact to spectral features, coherence, couplings and other properties. Indeed, we are absolutely convinced that this presents a leap forward for the quantum science community. We clarified these aspects in the revised version. Of course, any technique can only characterize the coherence of a system to a degree, to which the measurement process itself is not the dominating decoherence source. In this sense, it is indeed critical to explicitly discriminate our novel method from previously existing ESR-STM. The referee's criticism on our comparison we will address further below. Similarly, the referee's point on spatial resolution will be discussed along the detailed points further below.

We tried to allude to the aforementioned line of reasoning concerning the general motivation already in the first sentence of our manuscript, but we understand now that in the previous version this was too compressed to be intelligible. Thanks to the referee's comment we realized that our manuscript was lacking many necessary details. We also agree with the referee that the manuscript needed improvements in terms of a scholarly presentation for a broad readership. We hope to have resolved both shortcomings by adding all the relevant details, which are discussed along the individual points raised by both referees and that are discussed in the following.

Action:

We clarified at various locations in the manuscript that the aim is not constructing a quantum device but to add to the fundamental microscopic understanding of decoherence. We added a new dataset on PTCDA molecules (see new Fig. 2), the triplet-ESR properties of which was previously not known, to best of our knowledge. Other actions are listed along the individual points in the following.

R1: Below, I also include detailed scientific comments and questions that should be addressed in any future revision.

We thank the referee for the many detailed points that helped to improve our manuscript. To structure our reply, we re-ordered the referee's points according to four different categories: A) Claims and their justification, B) Comparison to ESR-STM, C) scholarly presentation and D) providing more details.

A) Claims and their justification

1.

R1: 115MHz detection corresponds nearly to an equilibrium spin temperature of mK. Why can this be detected nearly three orders of magnitude higher in temperature? For example, I believe all triple states are populated after excitation. But, isn't there a thermal probability to repopulate all triplet states after one of them decays?

"This sharp onset corresponds to a sub-nanoelectronvolt resolution in energy, enabled by eliminating main decoherence sources."

I do not think one can claim nanoelectron volt resolution in a non-equilibrium experiment, as this may violate thermodynamic arguments, especially asking the question about what temperature means in this system.

Authors: *The referee raised a very important point. There are indeed spectroscopic methods, in which the energy resolution is directly limited by temperature, but for others the spectroscopic resolution is independent of temperature. To facilitate the discussion, one may discriminate between the detection process and the system properties.*

It depends on the mechanism of the energy selectivity, whether or not the energy resolution of the detection is tied to temperature or not. For example, it is a well-known fact that Raman spectroscopy can provide sub-millielectronvolt resolution at room temperature (Nanoscale, 8, 6435 (2016)) without violating any thermodynamic laws. Another similar example is optical spectroscopy of NV centers. These techniques have in common, that the energy selectivity is given by the ability to drive a transition at well-defined photon frequency. Indeed, also in our case, the RF-induced transition can be viewed as the absorption of a photon with a well-defined energy.

In sharp contrast, in conduction spectroscopy the Pauli exclusion principle and the Fermi distribution act together as an energy filter, providing a selectivity as sharp as the Fermi distribution allows for. Hence, typical STM-based differential-conductance spectroscopy is always limited in energy resolution by temperature as implied by the referee. The same applies to any spectroscopy deriving from differential-conductance spectroscopy, for example, inelastic electron tunneling spectroscopy.

Whereas in ESR-STM a transition is resonantly driven, allowing – in principle – for a resolution independent of temperature, the existing ESR-STM implementations mostly relied on a disparity in the thermal population of closely lying states, such that experiments had to be conducted at very low temperatures to observe a signal.

In our implementation of ESR-AFM, all three triplet states that are being probed are almost an electronvolt above the electronic ground state. Hence, this is deeply out-of-equilibrium. We would like to add, that the out-of-equilibrium situation in our experiment is re-initialized with every pulse sequence and that the thermal equilibration between the different triplet states is not faster than their decay rates. Hence, thermal equilibrium is restored only after each cycle.

With a background in solid-state physics, one may be concerned that temperature would broaden the system properties irrespective of the detection method. However, an electrically isolated molecule has discrete energy levels instead of a quasi-continuum of states. The energies of the electronic states can be way sharper than $k_B T$. A molecule in an electronically excited state does not challenge the concept of temperature. The temperature in these experiments remains well defined by all other degrees of freedom that are still in thermal equilibrium. Note that, even in the ground state the electronic system of pentacene would not be well suited to define the temperature, since the lowest lying electronically excited state is ~ 1 eV higher in energy and not occupied to a measurable extent at 8 K.

2.

R1: I find this statement a bit unscientific in the course of reading the paper: “To demonstrate the enhanced coherence that results from our minimally perturbing detection scheme, we measured Rabi oscillations.” 1) I don’t know what enhanced coherence means. 2) To make this statement, a) the decoherence mechanism needs to be known, b) the “unperturbed” coherence time of that state needs to be known, and then shown that this method does not affect it.

Authors: *In the context of temporal coherence an enhanced coherence simply means a longer coherence time; we changed the phrasing accordingly. As stated above, we did not claim that our method does not introduce any decoherence, however, we do claim that the decoherence introduced by ESR-AFM is much smaller than in existing ESR-STM studies (see above) and we actually prove that it is not dominating in the case of protonated pentacene. We thank the referee for making us aware that our previous manuscript was unclear in this respect. We have strong evidence that the observed coherence time for the case of protonated pentacene is limited by the fluctuating nuclear-spin configurations. We believe that the remaining decoherence in deuterated pentacene is still dominated by fluctuating nuclear-spin configurations.*

Actions:

We improved the corresponding discussion in the manuscript.

3.

R1: “Further, the thick NaCl film completely decouples the molecule from the conducting substrate, eliminating decoherence caused by scattering electrons from the substrate^{11,18,31}” Again, I find this statement too strong. Indeed, there’s no electron transport, but the molecule is not completely decoupled from the film, as it is a dielectric environment. The authors already site an example of where the results differ from previous pentacene measurements. Also, the underlying film is ionic.

Authors: *In this sentence we referred to the conducting substrate, which in our case is the gold microstrip, and not to the ionic substrate as assumed by the referee. Indeed, in the second half of the sentence we did not repeat the word “conducting”, which may have led to the confusion. In the revised version we rephrased this sentence altogether to avoid any ambiguity.*

We feel that this consideration is important to discriminate our method from ESR-STM, in which a finite conduction to the substrate is needed.

Actions:

We changed the sentence as indicated.

4.

R1: “This reduction is due to the smaller hyperfine interaction of deuterium; the width is now dominated by nuclear electric quadrupole interaction.” This is an example of a suggestive statement where there is not direct experimental evidence: the authors do not measure this hyperfine interaction. I would tone down such statements, e.g. this reduction may be due...

Authors: *Yes, our assignment was based on previous literature (cited in this context) and is not proven in our manuscript. We followed the referee’s suggestion in toning it down.*

Action:

We toned down the sentence.

B) Comparison to ESR-STM

5.

R1: “The left flank of the signal corresponds to a broadening of 0.13 MHz (full width at half maximum, see Methods), being strongly reduced compared to ESR-STM” I found the comparison to ESR-STM apples to oranges. I think a comparison to ESRSTM can only be made if both methods are applied on one and same sample system. As the mechanisms are quite different, I do not find it appropriate to make direct comparisons which are better/worse.

Authors: *First of all, we would like to explicitly point out that we do not claim that our novel method would not introduce any decoherence at all. We will come back to that further below. Further, we do not intend to bash ESR-STM in any way. Instead, the ESR-STM community itself figured out the main decoherence sources of ESR-STM, namely (i) scattering from the tunneling electrons, (ii) scattering from the substrate’s conduction electrons and (iii) a fluctuating magnetic stray field from the tip. It is also the ESR-STM community that seems to be convinced that these decoherence sources limit the coherences observed so far; see references 11, 16 and 17. In addition, several ESR-STM groups have communicated to us that they actively tried to achieve longer coherences times and finally believe that the detection mechanism is being the limiting factor.*

If it is indeed true that the detection method of ESR-STM limits the observable coherence times as it is claimed by the ESR-STM community itself, the achievable coherence time is truly an important benchmark for a novel atomically resolved ESR technique.

Apart from other fundamental differences to ESR-STM, our work presents decoherence times that are approximately two orders of magnitude longer than what has been achieved before, which enables understanding decoherence on the atomic scale in the future as outlined further above.

However, in hindsight we fully agree with the referee that our manuscript gains by putting less emphasis on this comparison and rather concentrating on the multiple aspects, in which ESR-AFM is fundamentally novel.

Actions:

We removed emphasis from the comparison to ESR-STM and rephrased the remaining statements to make the above clearer.

6.

R1: Likewise, if there is no spatial resolution here, this is clearly an advantage of a method like ESR-STM.

Do the ESR spectra depend on where the probe is placed, with respect to the molecule or distance from the molecule? What does the imaging at the apparent measurement conditions look like?

Authors: One needs to discriminate two different types of resolutions. A non-invasive method can – at best – demonstrate resolution of a certain signal at the length scale of the signal's spatial variations. For the measurements presented in our manuscript, the peculiar resonance signal is a property of the molecule's electronic structure. The HOMO and LUMO are delocalized over the entire molecule and it is therefore natural to expect that the signal does not change as long as the tip is positioned above the molecule. As our method involves electron tunneling between tip and molecule (*Nature* 566, 245 (2019)), this sets the resolution basically the same as in STM.

In contrast, if the probe perturbs the system under study, another type of resolution can be achieved, which is often dubbed "scanning-gate microscopy" (*Science* 289, 2323 (2000)). The resolution in ESR-STM is typically of this latter type. While such resolution can provide extremely useful information, it comes at the cost of the probe being perturbative. This goes along with the community's observation that the method is limiting the observable coherence time. ESR-AFM with a suitable tip-termination may also provide such a scanning-gate-type resolution in the future and we are currently working on that. We currently see tiny shifts depending on the position of the probe above the molecule, which still needs to be investigated in detail. However, this goes way beyond the scope of our current manuscript. In case the tip is placed next to the molecule, the ESR signal in our spectra completely disappears.

Action:

We added ESR spectra above and next to the molecule as an Extended Data Figure 9.

7.

R1: At the end of page 5 and top of page 6, the authors point out decoherence mechanisms of other methods, but nothing about their own. Unless the authors want to claim that this is a perfect unperturbative method, then there also needs to be some text describing potential sources of decoherence from the measurement method, as this is the first paper.

- For example, if the tip itself has an electric dipole moment, it certainly can lead to dephasing of the molecule and potentially decoherence, right? Or are all probes prepared in the same way and have no effect?
- Likewise, the authors should also check other potential sources: Brownian noise of an oscillator for example (or amplitude/tip termination dependence). I understand what the authors want to claim here, but this is too strong of a statement.
- In the same way, I think this sentence could be a bit softer as it's a suggestion: "The long coherence time observed here is close to this intrinsic limit. This strongly suggests that decoherence due to the currentless but force-detected ESR, as introduced here, is negligible. Thus, ESR-AFM may even allow for coherent electron spin manipulation and read out at millisecond time scales^{24,32}, while providing atomic-scale real-space control^{22,33,34}"

Authors: We do not claim that our measurement does not introduce any decoherence at all. To avoid any such misunderstanding, in our revised version we follow the referee's advice and now explicitly mention one remaining source for decoherence, namely scattering with conduction electrons in the tip.

The strongest and clearest way to prove beyond speculation that the method is not the dominating source of decoherence for a given system, is to show that for another system, the

same method under the same conditions can be used to observe even longer coherence times. In fact, this we do in our work for protonated pentacene by comparing it to deuterated pentacene. We rephrased the corresponding paragraph to clarify this important point. We do not expect the electric dipole of the tip to play an important role in decoherence and we tested this by comparing different probe distances and cantilever amplitudes and include this information in the revised version.

Actions:

We follow the referee's advice and now explicitly mention one remaining source for decoherence. In the revised version we sharpened our line of reasoning and softened the sentences mentioned by the referee. We also added Rabi oscillations measured at different probe distances and cantilever amplitudes, see Extended Data Figure 7. They show no appreciable differences.

C) Scholarly presentation

We sincerely thank the referee for bringing up many points along which we could improve the scholarly presentation of the subject matter.

8.

R1: I had a hard time understanding what is actually measured, i.e. what is df_{norm} . Are the curves in e.g. Fig. 2a, what is directly measured in the df signal? If this is directly measured, then why would the force change in resonance? Or is this indirectly measured via this charge state, and thus extracted?

Authors: We thank the reviewer for bringing this to our attention. In brief, the triplet population is read out via the charge state. At resonance, the triplet population is reduced due to the shorter overall triplet lifetime of the molecule. Hence, in fewer cases, the pentacene is found in a triplet, which would be transferred to the cationic state. The ratio of cationic versus neutral encounters changes and this affects the Δf signal via electrostatic forces. Therefore, the Δf signal indirectly reflects the triplet population via the charge state (more details can be found in the methods section).

We normalize the Δf signal, because how exactly the charge state affects the Δf signal depends strongly on the tip height, the tip's work function, its electric dipole and other quantities. Rescaling the Δf signal with respect to the two involved charge states greatly reduces these dependencies and therefore renders a quantity that varies much less from tip to tip and relates much more straightforwardly to the triplet population.

Nonetheless, the Δf signal before normalization also shows the resonances, just with different scaling.

Action:

We extended the paragraph explaining how the Δf signal is linked to the triplet population.

We added the Δf signal before normalization as Extended Data Figure 2c and illustrate data propagation towards the normalized Δf signal.

9.

R1: Statement about Fig. 2A being related to hyperfine is not clear to me: does this lineshape depend on applied power/amplitude? Can this be better explained (or backed up by some experimental evidence)?

Authors: We agree with the referee that this is a nice opportunity to explain the cause of the line shape to a broad readership. The asymmetric line shape comes indeed from the hyperfine interaction, which we now explain in more detail in the methods section. In addition, above a certain threshold power, the signal is subject to power broadening. The data shown in the manuscript was recorded in the regime where the power broadening was minimized.

Action:

In the methods section we added two entire sections and a schematic figure (Extended Data Figure 3) to explain at length the situation of zero-field splitting as well as the reason of the asymmetric line shape. To the main text we added a few sentences referring to these

explanations. Furthermore, we added a figure on power broadening as Extended Data Figure 4.

10.

R1: What are the crystal field and spin Hamiltonians? What creates the zero-field splitting and why is there a difference between X, Y?

Authors: Here, it is not the crystal field but the spatial anisotropy of the molecule and its frontier orbitals that gives rise to the zero-field splitting of the triplet states. Roughly speaking, the localization of the two dipole-coupled electrons being different along the three high symmetry axes of the molecule gives rise to the differences between X, Y and Z (for details see the revised methods section).

Action:

In the revised methods section we provide the spin Hamiltonian and explain the origin of the zero-field splitting. The latter we also visualize in the newly added Extended Data Figure 3.

11.

R1: "The shift of the T_X - T_Z transition frequency by 60 MHz (~ 4% of its value) with respect to pentacene molecules in a host matrix^{24,25} can be rationalized by the different environments. In fact, the adsorption on a surface is expected to affect the triplet state electronic distribution along the z-direction (perpendicular to the surface), and thereby the energy of T_Z ." This was confusing to me. What give rise to the variations? What is the role of the ionic environment of NaCl, and may this play a role, similar to what is claimed in ref. 9?

Authors: We thank the reviewer for this comment. We realized that the second of the above sentences is not at all self-explanatory. Furthermore, when reading it in the context of the first sentence it might be even misleading. We, therefore, decided to remove the second sentence.

To nonetheless provide an answer to the reviewer's questions: The underlying NaCl may well affect the molecule's HOMO and LUMO densities and thereby affect the spin Hamiltonian via these modified densities.

Action:

We removed the second of the above sentences.

12.

R1: Can the two states be indicated in Fig. 2c? Not clear what the two states on the Bloch sphere are here, as there are multiple excitations. A picture and the relevant states would be useful.

Authors: We thank the reviewer for the helpful suggestion. For Rabi oscillations from driving the T_X - T_Z transition the predominant contribution at $t = 0$ is T_Z , then starting to oscillate towards a predominant contribution of T_X and back, which is indicated in the revised version. We added text to multiple places in the manuscript, but after thorough consideration we refrain from representing the spin system in a picture, because we are afraid that would cause more confusion.

Action:

We indicate the two states in Fig. 2c.

In the methods section we provide more details on the nature of the T_X , T_Y and T_Z states.

13.

R1: The simulations in Extended Fig. 3 are quite confusing. How is this calculated? Also, how are the occupations known, what Hamiltonian is expected and how is the effect of pumping considered in the problem? Is thermal broadening considered in this calculation?

Authors: We fully agree with the referee that instead of just mentioning the Maxwell-Bloch equations that we used to perform the simulation, we should be way more explicit. While we added all the details in the methods section, here we would like to keep it brief:

The Maxwell-Bloch equations are well established equations in the rotating frame approximation to describe transitions driven by an electro-magnetic wave. The initial

occupation of the three triplet states at $t_D=0$ is assumed to be equal, because of the non-spin-polarized preparation of the state (this is confirmed by our previous work (ref. 20). The states' lifetimes are known from the initial lifetime measurements. As the molecule is electrically isolated from the environment, it has discrete electronic states. Thus 'thermal broadening' will manifest itself in decoherence, which is included as such in the simulations.

Action:

We added all simulation details to the methods section, paragraph "Rabi oscillations". We added a new paragraph "Spin Hamiltonian and eigenstates" to the methods section.

14.

R1: I also do not fully understand what is shown in Extended Fig 4?

Authors: We thank the reviewer for noting that this figure is not clear enough. Extended Data Fig. 4 was essentially a sanity check for the important input parameters for the simulation of the Rabi oscillations.

Action:

As this figure was not essential and with all the additional data we ran out of space, we removed Extended Data Fig. 4.

15.

R1: "Similarly, the Rabi oscillations represent a time average of fluctuating nuclear spin configurations and, consequently, of a fluctuating Rabi frequency. Hence, the observed decay of the oscillations³ is likely dominated by the dephasing from the fluctuating resonance frequency, limiting the coherence time²⁹. Reducing the hyperfine interaction should then increase the coherence time further." Is this contradictory, or is it the statement that exchange with the hyperfine moments create dephasing? The way it reads now, it sounds like the resonance frequency is not cleanly applied.

Authors: It is indeed the statement that the interaction with the hyperfine moments creates dephasing. The resonance frequency (= frequency at which the molecule is in resonance) is something different than the applied frequency (= frequency from the RF source). Thus, the measured Rabi oscillations represent a time average of the instantaneous Rabi oscillations of the molecule in its different nuclear spin configurations. Since the molecule is then not always in resonance with the fixed applied RF field, the instantaneous Rabi frequency will vary.

Action:

We rephrased the sentences accordingly.

16.

R1: Much of this paper assumes the reader has digested the group's previous Science paper. The text would benefit a lot from some better illustration and clearer descriptions. Can these charge states be labeled in Fig. 1, and where the unpaired spins are?

Authors: We thank the reviewer for noting that the manuscript's understanding should not rely too much on the group's previous Science paper.

Action:

To address this issue, we:

- Refer to the already existing Extended Data Fig. 1 from the main text, because it illustrates the detection scheme used in the previous Science paper.
- Most importantly, during all spin manipulation the molecule is always in the neutral state. The charge state is only relevant for the read-out. We, therefore, elaborate on the latter in the methods section and in Extended Data Fig. 1.
- We added the unpaired spins to the molecule in Fig. 1

D) providing more details

18.

R1: The authors do not provide any imaging and details about the NaCl growth. Are there defects, is it polycrystalline? Will the quality of the NaCl (or lack of it), lead to variations between measured molecules (e.g. due to defects)?

How many tips/molecules were measured, are they all the same, or are there variations or tips that doesn't show resonance? Does every probe show a resonance? How are the probes created? These are important details when benchmarking a new method.

Authors:

Thanks for pointing this out. Locally, the film is crystalline and there are no defects except for step edges, as is now also confirmed by the new high resolution image displayed in Fig. 4. We added ESR spectra for two molecules in Fig 4, the shape of the ESR signals are identical within the uncertainty margin, and their resonance frequencies are almost identical despite the difference in their environment.

In case of pentacene, the T_X - T_Z transition was measured for 19 individual pentacene- h_{14} molecules, 20 pentacene- d_{14} and 1 pentacene- hd_{13} ; for 18 of these molecules we also measured the T_X - T_Y transition. The T_Y - T_Z transition of PTGDA was measured for 2 molecules. In total 16 different tips were used for these measurements.

Basically, for every tip that showed a large enough signal change upon charging a molecule, we observed an ESR resonance, only once this was not the case.

Action:

We added the statistical information and a table with resonance frequencies for multiple molecules (Extended Data Table 1) to the methods section, as well as more information on the growth parameters and tip preparation.

19.

R1: The way the striplines are designed, are there field gradients and certain selection rules to be concerned with? What is the actual direction of the AC field with respect to the molecule? Does the molecular orientation matter?

Authors: *By design, no large field gradients are expected. There are indeed selection rules with respect to the alignment of the molecules relative to the field. We added the selection rules to the methods section as well as a description of the required magnetic field components for driving the observed transitions.*

In brief: since the field lines loop around the microstrip, a magnetic field in the xy plane perpendicular to the strip direction is expected. In addition, the inductive coupling of the signal into the microstrip adds a vertical component to the field. The orientation of the molecules does thus indeed matter, since for instance only the y-component of the B-field drives the T_X - T_Z transition.

This can be tested by relating the molecular orientation in the images to the direction of the RF field, which in turn is determined by the sample setup. Interestingly, the z-direction of all molecules (roughly) coincides, while the x- and y-directions vary, providing a nice testbed for these considerations.

Maybe the referee wanted to implicitly suggest this as a convincing additional demonstration of the direct microscopic understanding that can be provided by the new method. We therefore ran a new measurement campaign on this topic that is included in the revised manuscript in Fig. 4 (and Extended Data Fig. 8).

Action:

Selection rules and a field description are added to the methods section. An entire new dataset was measured and included in the revised manuscript.

20.

R1: How is the transmission of the stripline calibrated?

Authors: *We characterized the transmission function from the RF-signal feedthrough at the cryostat to the single-loop coil that inductively couples the signal into the sample by measuring the magnetic field at the coil. This transmission function varies appreciably with frequency over the frequency range from 0.1 to 2.0 GHz, but only smoothly. As the*

transmission function is smooth it barely varies ($\pm 4\%$) in the frequency interval of 17 MHz around the T_x - T_z transition.

The sample microstrip structure is not expected to introduce sharp resonances, since the wavelength at the relevant frequencies is much larger than the entire structure. This assumption is supported by an ex-situ spectrum-analyzer measurement.

We assume that the reviewer's question is motivated by the importance of the transmission function in ESR-STM. We note, therefore, that in our implementation of ESR-AFM the RF-signal creates the magnetic field in a direct way, instead of adding to the bias voltage that then acts indirectly as an effective magnetic field.

Hence, the main concern connected to the transmission function in our case is to assure that variations in the transmission function do not appreciably distort the line shape of the resonances. Compelling evidence for the total transmission function not significantly affecting the line shape comes from the observation of peaks that are shifted in energy but of identical shape for different molecules.

Action:

We extended the previously existing description of how the transmission function was determined.

21.

R1: What determines the error bars in the df signal in all this data? Are these error bars only for a single measurement, and all are otherwise the same, i.e. it's extremely reproducible from molecule to molecule?

Authors: As stated in the methods section, all data traces were repeated a number of times, and the error bars were extracted as the standard deviation of the mean of these repetitions. Thus, irrespective of its source, any type of non-systematic error will enter the error bars. Indeed, we did this procedure for every individual data point independently. The error bars are reproducible from molecule to molecule. As can be expected, probe tips that give a strong response to charging (a large charging step in the Kelvin parabola) provide a better signal-to-noise ratio and therefore a smaller relative error bar.

Irrespective of how the error bars are determined experimentally and how reproducible they are, we analyzed in detail how different physical contributions may enter the resulting error bar. The three main sources of uncertainty are the statistical error from the finite number of repeats, the remaining drift of the tip height and the noise on the frequency shift. Depending on the experimental conditions, either of these three sources dominates.

Action:

We added a brief discussion of the main sources of uncertainty in the methods section.

23.

R1: "This splitting arises predominantly from the dipole-dipole interaction of the two unpaired electron spins. It is therefore governed by the spatial distribution of the triplet state and can serve as a fingerprint of the molecule." Likewise, won't this lead to a distortion of the molecule what the AFM may not see but can be learned about from the spectral data? Also when it is charged?

Authors: We thank the reviewer for pointing out that the ESR resonance frequencies provide information that the AFM is not sensitive to. In particular, it provides excited-state properties, which is notoriously difficult to obtain by AFM. We note that the ESR spectra presented here refer to the molecule in its neutral state only. The charged state is only used as vehicle to read out the signal.

Action:

We clarified that the ESR spectra relate to the neutral molecule.

24.

R1: Does the Q factor of the oscillator change during the measurement, or is there any measurable dissipation/excitation signal?

Authors: We thank the reviewer for this detailed question about the observed signal. We did not observe any significant dissipation/excitation signal (change in the Q factor) during an RF frequency sweep (see below). Note that the relative fluctuations in the dissipation are on the 10^{-3} scale.

Action:

We added a sentence with this information to the methods section.

25.

R1: In 1b, what does a broadband pulse mean, isn't this continuous wave excitation?

Authors:

The term broadband does not refer to the duration of the pulse. Instead, the pulse consisted of frequency components over a broad range. It was deliberately designed this way in frequency space.

Action:

We specified the nature of the signal in more detail.

Miscellaneous:

26.

R1: Refs. 15-16 in the abstract are misleadingly cited, none of this work is related to a quantum device.

Authors: References 15 and 16 are indeed references to atomic manipulation by means of STM in general and were therefore cited after the words 'Atomic manipulation' at the beginning of a sentence.

Action:

To avoid any confusion, we removed these two citations.

Referee #2

It is an impressive surprise that it is possible to observe single-molecule spin resonance in this force-detection scheme, where no net tunnel current, magnetic tip, single-electron transistor, or optical signals are used. In this way, this work is groundbreaking. However, it is not clear how widely applicable the technique may be, and will be quite challenging for non-specialists to follow the main physics and techniques involved, in order to imagine its use outside this specialized system. Since it does not appear to show important new properties of the molecular spin studied, this is primarily a paper about the new technique, and would need to characterize the technique in more depth and show that it applies to a variety of other systems of possible interest. My assessment is this paper surely does belong in a high-quality journal but does not meet the very high level of broad accessibility or impactful applicability that is required for a Nature publication. In any case, the following comments and questions may help to improve the manuscript.

Authors: *We thank the referee for the appreciation of our work. We hope that with our improvements of the manuscript we have addressed the main criticism, namely, the accessibility to non-specialists and the demonstration of a wider applicability.*

1.

R2: It takes quite a bit of effort in studying this paper to feel confidence that the observed Rabi signals cannot be due to some other oscillatory property, such as an electronic oscillation, mechanical mode of the cantilever, or of surface acoustic waves in the Au microstrip layer, as seen for example in the MHz surface acoustic waves in gold films by [Sheng et al. PRL 129, 043001 (2022)]. After examining the data that is shown, I am quite convinced that ESR and Rabi oscillation is observed, but it would be more immediately persuasive if it were shown that the Rabi frequency is proportional to the RF amplitude. I encourage the authors to include that measurement if it is available. Such a measurement would also serve as a basic control and characterization test that many readers would expect to see for this scheme.

Authors: *We thank the reviewer for the great idea to show that the Rabi frequency is proportional to the RF amplitude.*

Action:

We added this requested data to the manuscript as Extended Data Figure 6

2.

R2: Please give an idea of how broad the data set is. For how many molecules was spin resonance observed for each isotopologue?

Authors:

In case of pentacene, the T_X - T_Z transition was measured for 19 individual pentacene- h_{14} molecules, 20 pentacene- d_{14} (of which one with ^{13}C nucleus) and 1 pentacene- hd_{13} ; for 18 of these molecules we also measured the T_X - T_Y transition. The T_Y - T_Z transition of PTCDA was measured for 2 molecules. In total 16 different tips were used for these measurements.

Action:

We added such information to the methods section.

3.

R2: The reduction in line width when using deuterated molecules is compelling, and is compatible with previous ensemble ESR. However, it is confusing why the h-d13 line here seems to be essentially the same width (~3 MHz) as for h14. How is this possible? How is it determined that the molecule is an instance of h-d13 rather than an errant h14 found due to contamination from previous doses? Is there any measurement besides the line width that supports assignment of the isotopologues?

Authors: We agree that at first glance it is very surprising that a single hydrogen in an otherwise deuterated molecule gives rise to a broadening that is similarly wide as for a completely protonated molecule.

The assignment to the isotopologue is done in comparison to previous work, based on the line shape, which does not only include the width but also the shape (more symmetric).

The shape is clearly incompatible with a fully protonated pentacene molecule.

Since it is a second order hyperfine interaction, the mere presence of one nucleus with strong hyperfine interaction can give rise to a substantial line broadening, as seen from the simplified formula describing this interaction (analogous to ref. 27):

$$E_{HF}^2 = \left[\left(\sum_{i=1}^{13} \pm E_{D_i} \right) \pm E_H \right]^2 = \left(\sum_{i=1}^{13} \pm E_{D_i} \right)^2 \pm 2 \left(\sum_{i=1}^{13} \pm E_{D_i} \right) E_H + E_H^2$$

As can be seen from the second term to the right of the equation, the one proton also affects how all the deuterons affect the line, thereby also changing the overall shape of the line. A more depictive reasoning is as follows: while many nuclei with similar interaction strengths can partially compensate each other because of their random spin orientation, one strongly interacting nuclear spin cannot be compensated by others. We note that it also depends on the proton's position, how strong its nucleus acts on the electron spins.

Thanks to the referee's comment we reconsulted the literature, this time also considering the possibility of a ^{13}C instead of just concentration on hydrogen isotopes. In fact, for our previous case, we find an even better match to existing data, in which one of the pentacene's carbon was a ^{13}C .

In the revised version we replaced most of the data by new data, to fulfill the referee's request for better image quality. In this new dataset we found another molecule with a line shape that is indicative to its isotope composition as pentacene- $h\text{-}d_{13}$. However, this time we phrase the assignment more tentatively.

Action:

We added this reasoning to the methods section. We added power dependent spectra to illustrate that the different line shape is not due to power broadening, see Extended Data Figure 4.

4.

R2: It is not clear how broadly applicable the technique can be. The triplet excited state in which splittings are set by the molecular anisotropy rather than by Zeeman energies. The pentacene triplet states are a well investigated and historically important model spin system for previous single-molecule ESR, but it would seem to be a fairly special case. How broad a range of molecular spins have accessible triplet states, long enough state lifetimes, sufficiently distinct relaxation times, and nearby states for spin-to-charge conversion, all of which are required here? Alternatively, is there any clear path to adapting this method to other kinds of molecular or atomic spins?

Authors: We thank the reviewer for the in-depth comment. The reviewer is correctly mentioning the most important requirements. For not too small molecules featuring a delocalized electron system, the triplet states should be within the energy range of our experiment. In organic compounds the spin relaxation time tends to be large because of low spin-orbit coupling, pentacene is not an exception in this respect. Similarly, the distinct relaxation times seem not uncommon (*ChemPhysChem*, 18, 6-16 (2017)) for not too small molecules featuring a delocalized electron system. In addition, other groups have demonstrated that all-electronic pump-probe STM experiments down to nanosecond timescales are technically feasible (*Applied Physics Letters*, 103, 183108 (2013)), such that even lifetimes in the ns range would suffice in principle. Within the constraints of not too small molecules featuring a delocalized electron system as mentioned above, we would always expect nearby charge states within the bandgap of the NaCl. Note that, the level alignment between the charged ground state and the neutral ground state needed for the spin-to-charge conversion is tuned by the gate voltage.

Hence, while we do not claim that our method is applicable to any type of molecule, the class of molecules to which it could be applied seems rather large. We would like to emphasize that we believe that one of the future potentials of our technique lies in correlating spin coherence to local atomic information and thereby facilitating an atomistic understanding of decoherence processes. To this end, one would choose model systems exhibiting long spin lifetimes – and such systems typically fulfill the above requirements.

Another scientific application lies in quantum sensing, that is, using the molecule under study to sense other species in the environment. In such an application, being limited to a few sensor molecules does not pose a severe restriction.

Finally, we do see clear paths to adapting this method to broaden its applicability. For example, combining our method with spin-polarized tunneling, would enable to initialize a difference in triplet state populations and access otherwise non-accessible triplet state transitions.

While we believe that this extensive discussion is outside the scope of this article, we do agree that it is important to demonstrate to the reader that this technique is certainly not limited to pentacene.

Action:

To demonstrate that the applicability to pentacene is not unique, we studied PTCDAs as a second molecule and added the corresponding data in Fig. 2.

5.

R2: The usual spin-1/2 ESR is sensitive directly to B fields and couplings in well established ways, but the pentacene transitions are insensitive to B fields to first order, giving them good coherence but making them poor as sensors. On the other hand, as qubits for quantum information they would need to be coupled together and sensitive to their neighbors. Please mention how these could plausibly be applied as detectors, or as long-coherence coupled qubits, or in some other way.

Authors: The referee's comment is very useful to sharpen our line of reasoning. The reviewer is right to state that the pentacene's zero-field splitting is insensitive to B fields to first order.

Nonetheless, pentacene can be used as a detector, as we see small shifts of the ESR line depending on the environment. These shifts are appreciably larger than the energy resolution of the experiment (see also the newly added Extended Data Table 1).

Moreover, the relaxation time is very sensitive to the spins in the environment and can be used to probe this.

Importantly, one is not restricted to zero-field splitting as used here: a static magnetic field can easily overcome the zero-field splitting and thereby give rise to a splitting into T_0 , T_+ and T_- states. These states are sensitive to B fields in first order, increasing the coupling to neighboring pentacene molecules as well as to the environment.

Going beyond that, imagine a local magnetic moment that can be switched on and off, for example by controlling the charge state of an adsorbate (refs. 44,45) or in a spin crossover compound (ref. 43). This way, one could deliberately switch between two sets of eigenstates of the system (T_x , T_y , T_z vs. T_0 , T_+ , T_-) as well as switching on and off the coupling between neighboring spin systems.

Action:

We clarified in the introduction that the aim is not constructing a quantum device but to add to the fundamental microscopic understating of decoherence. We added some of the above aspects to our manuscript to strengthen the outlook on quantum sensing capabilities.

6.

R2: A description of the quantum states T_x , T_y , and T_z is needed, along with the Hamiltonian leading to them and the resulting selection rules for coherently driving transitions among them given the in-plane RF magnetic field.

Authors: We fully agree.

Action:

We added the requested description with a physical discussion to the methods section. Among other measures we added a new paragraph "Spin Hamiltonian and eigenstates" and Extended Data Figure 3 to the methods section.

7.

R2: What model was used for the Rabi oscillation part of the Maxwell-Bloch simulations?

Authors: We used the Maxwell-Bloch equations to perform the simulation, which are well established equations in the rotating frame approximation to describe transitions driven by an electro-magnetic wave. We fully agree with the referee that instead of just mentioning that we used these equations, we should be way more explicit. We added, therefore, the relevant equations as well as the used parameters to the methods section.

Action:

We added all simulation details to the methods section, paragraph "Rabi oscillations".

8.

R2: Can the orientation of the molecules be deduced from the Rabi rates and does this agree with the imaged orientation?

Authors:

The referee is correct: Due to the selection rules for the three different transitions, for each of them, only the RF-field along one of the three high symmetry directions of the molecule matters. As the referee points out, this can be tested by relating the molecular orientation in the images to the direction of the RF field, which in turn is determined by the sample setup. Interestingly, the z-direction of all molecules (roughly) coincides, while the x- and y-directions vary, providing a nice testbed for these considerations. We agree with the referee that this is a convincing additional demonstration of the direct microscopic understanding that can be provided by the new method. We therefore ran a new measurement campaign on this topic that is included in the revised manuscript.

Action:

Entire new dataset measured and included in the revised manuscript in Fig. 4 and Extended Data Fig. 8.

9.

R2: Reference [25] (Kohler et al 1993) section 2.2 appears to be a good presentation of the states, and the authors of this paper could refer readers directly there, or even better, spell it out in their own terms. If I understand correctly from [25], the triplet states can be written (unnormalized, and using the surface-normal z basis) as $T_x = |11\rangle - |00\rangle$; $T_y = |11\rangle + |00\rangle$; and $T_z = |10\rangle + |01\rangle$. The driving field B_{RF} would need to create an oscillating energy difference between the sums and differences of the two states involved in a transition. For example, driving between T_x and T_y happens when the state $T_x + T_y = |11\rangle$ differs in energy from $T_x - T_y = |00\rangle$, and this indeed is the case for a B field in the z direction. However, the strip line appears to give an essentially xy-plane B field, so please explain how this transition is driven coherently. A similar analysis for the T_x to T_z transition is also needed.

Authors: At this point we would like to thank the referee explicitly for diving so deeply into the topic of our manuscript. The understanding of the reviewer is correct, and we agree that we should explain all this in the manuscript.

Action

We added the selection rules as derived from the zero-field Hamiltonian to the methods section. We included an explanation on how both transitions can be driven with our experimental setup.

10.

R2: The highly-asymmetric line shape for h14 pentacene is well established from previous publications, but for general readership it would help to provide a qualitative idea why the hyperfine coupling gives such asymmetry.

Authors: We fully agree.

Action:

We added the requested description with a physical discussion to the methods section. We also included a schematic to visualize the origin of the asymmetric line shape, see Extended Data Figure 3c.

11.

R2: In the only AFM image (inset greyscale image of Fig. 3c) of the topography, it is not very convincing that individual molecules are observed. It is helpful that previous work by this group on triplet quenching showed that they are capable of discerning the molecular orientation and atomic environment using AFM alone. It would strengthen this paper to show clearer images, and showing that the presence or absence of the ESR and Rabi signals correlates with presence of the discernable molecule and its orientation.

Authors: *We fully agree and we are grateful for the referee's trust in our capabilities.*

Action:

We measured and added the requested high-resolution images and ESR and Rabi signals, see Fig. 4 (and Fig. 2). Extended Data Fig. 9 is added to demonstrate that next to the molecule the ESR signal disappears.

12.

R2: In Ext. Data Fig. 2, should $\langle\Delta f_0\rangle$ be simply $\langle\Delta f\rangle$?

Authors: *We thank the reviewer for spotting the typo. Once more this demonstrates how meticulously the referee has reviewed our manuscript and we would like to express our gratitude for that.*

Action:

We corrected our mistake.

13.

R2:

The word "broadband" in Fig.1 caption is confusing. Perhaps it means "radio-frequency", at a single frequency

Authors: *The term broadband meant to express that the pulse consisted of frequency components over a broad range. It was deliberately designed this way in frequency space.*

Action:

We specified the nature of the signal in more detail.

Reviewer Reports on the First Revision:

Referee #1 (Remarks to the Author):

In this revision, the authors have provided a revision and a rebuttal to address the open points raised in the previous round. The largest changes include (a) more detailed explanation of what they did, (b) new data on a different molecule, namely PTCDA, to demonstrate that the method is not limited to one molecule. I appreciate that this revision involved a lot of work, and many of the points have greatly improved the readability of manuscript. However, I cannot recommend publication of this work for two reasons: (1) the authors have not properly addressed the main concerns raised before, which involve standard modest scientific discussion of basics concerning their method. They are crucial points to substantiate their scientific claims, based on learning about atomic scale environments and quantum sensing. I can appreciate that not everything is known or understood, but that does not excuse a proper discussion (or outlook), about crucial points related to method development. This is not written at a level I would deem worthy of a Nature publication. (2) Concerning the PTCDA, I find the data exciting and certainly a perspective for expanding the method to new materials. Yet, it is mentioned in all but one sentence, which is not appropriate for a manuscript. I absolutely cannot endorse publishing an experiment that was only measured twice, as mentioned by the authors. If this is correct, I was shocked that this is put into a paper, and that these statistics are not mentioned (e.g. only the benzene statistics are listed). While it is most likely that this is reproducible, proper statistics are essential before publishing any work.

As I do not want to engage in a long round of referee rebuttals, I would recommend rejecting this paper, unless the authors take the time to address the straightforward scientific points raised before. I have tried hard to find a constructive manner to endorse this paper, but in the end, I find it an exciting result embedded in a poorly written paper. Most of these points would involve an open scientific discussion in their paper and better use of scientific language. To be clear, I will summarize these points again below.

- The authors write that the strength of their method, in the first portion of the rebuttal is "It is the possibility to image and characterize the unique quantum system under consideration in all its atomic details." They further go on to discuss the atomic environment as a strength of their method. Yet, they cannot claim this with what is written:
 - o (i) Again, they do not provide any text about the spatial resolution of their method (not x,y nor z), which was one of the main points from the previous reply. Until this is discussed, this method should not be called ESR-AFM, as microscopy involves spatially dependent imaging. It is essential to discuss spatial resolution: 1) what is the distance dependence, and does the tip couple to the whole molecule or to some component of it? 2) Is the method capable of atomic resolution as they seem to want to claim, intramolecular resolution, or is it only sensitive to small height changes, and there's no lateral resolution? In their rebuttal, they do not answer my question (e.g. see point 6, and begin a discussion about something unrelated). This is textbook stuff related to microscopy and a new method development.
 - o (ii) They give zero discussion about what they can learn about how the environment changes the coherent behavior (e.g. do they see a difference between coherence times for different molecules, due to defects). This is the main claim of their paper, yet the paper only exhaustively wants to claim that they are studying a nearly free standing molecule and their method does not perturb the measurement. They want to claim there are no defects (which is not believable and even looks present in Fig. 2C), and that their method does not perturb the molecule. Ionic defects and step edges in their insulator can present significant potential variations that can change their level structure. Then they see a variance in their statistics, which they do not explain and need to account for (textbook error analysis). Why is there a change in their frequencies in extended table 1? There's no change between metal and CO tips, where the latter may even have a strong electric dipole. Does this mean that they are not sensitive to the atomic environment, which then a contradiction to what they claim? I would guess that the points "change the quantum system" as the environment is changed. If there are no effects there, then they cannot claim their method is

sensitive to any of this, and thus it is contradictory. This needs to be discussed and clarified, as it is very confusing.

- In the discussion about energy resolution, the authors do not address the underlying issue about what their energy resolution is, or if they can claim it. This was linked to the discussion about thermodynamics and non-equilibrium. They provide a lengthy and confusing discussion, some of which I believe is not completely correct, about thermodynamics. The original point is the word energy resolution cannot be used in the context they describe. To claim this is an energy resolution of the method, it must be determined against a known benchmark, and arguably done in equilibrium. This would illustrate that the energy resolution is inherent to the method. For example: is the energy resolution dependent on the molecule they probe, or the dynamics they excite? To be clear, temperature in this system should change intensities of these peaks, and one cannot compare this method with the ensemble measurements they try to argue with (e.g. where bosons are also studied). Here, the question is if what they claim as resolution is dependent on the molecule and their excitation conditions.

Often, the authors use words like isolated as they want to accentuate that they are not strongly coupled to an electron bath. But, this reads very contradictory in their paper. Isolated quantum systems are typically called closed quantum systems, and they are far from this limit. Also, even though they suppress electron scattering does not mean they can ignore decoherence from other sources (e.g. phonons, dynamically induced distortions of the molecule, tip/molecule interactions, other electric fields generated by defects, etc.). What is the role of Brownian motion of the tip? There's no satisfactory discussion about the limits of the method, and what sources of decoherence exist. Simply statements about what does not affect the measurement. In the tone of the manuscript, and the poor comparison to ESR-STM, they signal that they are non-invasively probing a closed quantum system.

- If it is correct that PTCDA was measured twice, it needs to be removed from the manuscript, or more properly studied. To standards in the field, this is not considered statistics and should not be published, unless it is clearly discussed, including the lack of statistics. If they want to include this molecule in the paper, they really need to better describe what they measure beyond one sentence.

Minor points:

- I found many instances of the word "tiny" in the manuscript, which is not scientific language.
- I think the introduction is overstated, for example "the construction of understanding of decoherence in complex solid-state quantum devices requires ultimately controlling the environment down to atomic scales..." A large part of the solid-state community in quantum technology focuses on superconductors. The length scale of interest is the coherence length, which is on the order of 10s of nms, and not focused on atomic-scale control."
- I would refrain from using words like isolated in context of quantum systems, as reasoned above.

Referee #2 (Remarks to the Author):

In the 2023 resubmission of "Atomic-scale access to long spin coherence in single molecules with scanning force microscopy" the authors have responded in detail to the comments by both referees. They have made many changes and included major new data that greatly strengthen the paper. The inclusion of the PTCDA molecule, the wide-frequency spectra of both molecules, and the clear correspondence between bond-resolved images and their ESR spectra, all acquired using the same tip, are an outstanding experimental accomplishment that supports the main claims. Other new information improves the paper markedly, such as the Rabi dependence on the drive amplitude, the acquisition statistics, the line shape explanation, the selection rules, and the discussion of Rabi mechanisms, rates, and molecular orientation. These changes strengthen the paper to the level that I recommend publication in Nature.

A plausible decoherence mechanism that was not mentioned earlier is the fluctuating Stark shift of the resonant frequency that is due to the uncontrolled electric field fluctuations that arise from uncontrolled motion of the conductive AFM tip. Please pardon my oversight in neglecting to discuss Stark shifts in the first evaluation. Decoherence is emphasized in this paper, so I recommend mentioning the Stark shift mechanism and providing an estimate of the range of plausible values for the line broadening, even if this discussion simply rules out observable Stark effects here. A rough estimate: at a bias voltage of 2.5 V during the dwell period, an effective junction thickness of ~ 2.5 nm gives an electric field of ~ 1 GV/m. If the unintended fluctuating tip motion is ~ 2.5 pm, then the electric field fluctuations are ~ 1 MV/m. A Stark shift of 1 Hz/(V/m) would give an ESR line broadening of ~ 1 MHz, which is in the range of the broadening observed. Stark shift sensitivities vary widely, and the authors may have better estimates of all these parameters given their specific spin system and experimental setup. The tunnel barrier includes the thick NaCl, but this has a large dielectric constant, so its vacuum-equivalent effective thickness is less. A determination of this Stark-shift sensitivity may already be available to the authors if they measured how much the frequency shifts with different dwell voltages or with significantly different NaCl thicknesses. Extended Fig. 7 is a step in that direction but doesn't vary V or vary z enough to estimate the effect. An indication that Stark shift may be negligible is the observation that the cantilever oscillation (0.1 nm zero-to-peak or peak-to-peak?) occurs on the same timescale as the Rabi RF pulse duration but there does not seem to be a dependence on the cantilever phase relative to the RF pulse. Do the authors synchronize the start of the RF pulse with the cantilever as they do for the charge state control, or is this unnecessary?

The following may optionally help to improve the clarity:

Line 135: In the new paragraph "The corresponding Rabi oscillations...": corresponding to what? Presumably this means deuterated pentacene.

Line 158: How is being "out-of-thermal-equilibrium" relevant to measuring at the conditions used? Does it mean that the initialized state is other than Boltzmann, or does it mean something else?

Line 161: "The increased spin coherence...": rephrase or state what it's compared to.

Line 282: "represented by the two arrows" perhaps refers to the tiny black arrows on the molecule added in this version. These arrows may represent spins but are illegible at this scale and are not needed.

Line 286: "Decay of T1": The authors are careful to distinguish the label T1 (any of the triplet states) from the variable tau (lifetimes), but T1 is used for relaxation times in many other publications, so this may confuse casual readers. Consider adding a few clarifying words like "decay of the triplet states"; and "the lifetime measurements of the triplet state T1" instead of "the T1 lifetime measurements".

Line 299: "perdeuterated pentacene-d14 (A) and PTCDA (B)" is unclear whether it is perdeuterated PTCDA.

Line 300: "Note that for PTCDA there is a tiny signal at 1501 MHz that is too small to be observed at the parameters used here": Did the authors observe this signal at some other parameters? If it is not shown anywhere in this paper, say something like "not shown".

Line 317: The variable tS is used before being defined, and what does the letter "s" stand for? The 45.1 microsecond delay mentioned in line 97 is presumably tS. The times tS and tD are illustrated in the inset of Fig. 3a but are too small to be legible; taking space for that diagram is worth it.

Line 319 "...as indicated": indicated where?

Extended Data Fig. 1: The yellow labels P+ and D0 are too light to be legible.

Line 801: "two per orientation": are there precisely two orientations observed?

Author Rebuttals to First Revision:

Referee #1 (Remarks to the Author):

In this revision, the authors have provided a revision and a rebuttal to address the open points raised in the previous round. The largest changes include (a) more detailed explanation of what they did, (b) new data on a different molecule, namely PTCDA, to demonstrate that the method is not limited to one molecule. I appreciate that this revision involved a lot of work, and many of the points have greatly improved the readability of manuscript. However, I cannot recommend publication of this work for two reasons: (1) the authors have not properly addressed the main concerns raised before, which involve standard modest scientific discussion of basics concerning their method. They are crucial points to substantiate their scientific claims, based on learning about atomic scale environments and quantum sensing. I can appreciate that not everything is known or understood, but that does not excuse a proper discussion (or outlook), about crucial points related to method development. This is not written at a level I would deem worthy of a Nature publication. (2) Concerning the PTCDA, I find the data exciting and certainly a perspective for expanding the method to new materials. Yet, it is mentioned in all but one sentence, which is not appropriate for a manuscript. I absolutely cannot endorse publishing an experiment that was only measured twice, as mentioned by the authors. If this is correct, I was shocked that this is put into a paper, and that these statistics are not mentioned (e.g. only the benzene statistics are listed). While it is most likely that this is reproducible, proper statistics are essential before publishing any work.

As I do not want to engage in a long round of referee rebuttals, I would recommend rejecting this paper, unless the authors take the time to address the straightforward scientific points raised before. I have tried hard to find a constructive manner to endorse this paper, but in the end, I find it an exciting result embedded in a poorly written paper. Most of these points would involve an open scientific discussion in their paper and better use of scientific language. To be clear, I will summarize these points again below.

Authors: *We regret that despite our effort, we did apparently not address the reviewers points to their satisfaction. It seems there are misunderstandings and that we apparently talked at cross-purposes. It even appears to us that we somehow upset the reviewer with our rebuttal, which was not at all our intention. Taking all points together, we understand that the referee's main irritation comes from an impression that we present our work not modest and open enough. In this spirit, we tried to revise our manuscript towards further toning certain things down.*

Having done so, we sincerely ask the reviewer to sympathetically look over our manuscript, whether with these changes we were able to resolve these irritations. Irrespectively, we reply to each of the points below.

Actions:

See detailed points below.

1. R1

The authors write that the strength of their method, in the first portion of the rebuttal is "It is the possibility to image and characterize the unique quantum system under consideration in all its atomic details." They further go on to discuss the atomic environment as a strength of their method. Yet, they cannot claim this with what is written:

(i) Again, they do not provide any text about the spatial resolution of their method (not x,y nor z), which was one of the main points from the previous reply. Until this is discussed, this method should not be called ESR-AFM, as microscopy involves spatially dependent imaging. It is essential to discuss spatial resolution: 1) what is the distance dependence, and does the tip couple to the whole molecule or to some component of it? 2) Is the method capable of atomic resolution as they seem to want to claim, intramolecular resolution, or is it only sensitive to small height changes, and there's no lateral resolution? In their rebuttal, they do

not answer my question (e.g. see point 6, and begin a discussion about something unrelated). This is textbook stuff related to microscopy and a new method development.

Authors: As we believe that we did answer the reviewer's question in point 6 of the rebuttal, as well as both points the reviewer raises again here, we must conclude that there is a misunderstanding. We assume that the referee expected us to not only address these points in the rebuttal, but also have it explicitly mentioned in the main manuscript itself. We also agree with the referee that this is important to the general reader. Therefore, we include this discussion now in the main text. We ask the referee for their understanding that due to the length restrictions this has to be very brief.

To directly answer to the referee's points also here: (ad 1) The resonance signal is related to the triplet in the molecule, being formed by one electron in each the HOMO and the LUMO. Here, HOMO and LUMO are delocalized over the entire molecule. For other compounds this might be different, and the triplet may be localized. To bring the molecule into the triplet state before each probe cycle, electron tunneling between the tip and the HOMO and the LUMO has to take place. Therefore, the spatial resolution of the ESR-AFM signal is predominantly determined by the distance dependence of these tunneling processes. This type of resolution enables to assign a given ESR spectrum unambiguously to one individual molecule beneath the tip. We explicitly demonstrate that the signal is absent if the tip is positioned laterally away from the molecule (Extended Data Fig. 9).

(ad 2) Above the molecule, where the formation of the triplet is possible, the presence of the tip will certainly influence the ESR resonance, for example by the Stark effect as mentioned by both referees. This might also occur indirectly, for example, the Stark effect might influence the electron distribution and thereby the splitting in T_x , T_y and T_z . We do see a Stark shift and this we will try to make use of in the future, but any details on that clearly go beyond the scope of the present work. However, we now mention the Stark shift and its possible use. (See also next point)

To summarize, so far, we clearly show that we can attribute the minute details of an ESR spectrum unambiguously to one individual molecule. The latter we can then image with the atomic-scale spatial resolution of AFM. We combine these two aspects in one instrument, with the same tip, for one given molecule.

This is what we actually claim, see second-last sentence of our conclusion:

'Directly resolving the atomic-scale geometry in relation to minute differences in zero-field splitting, spin-spin coupling, spin decoherence, as well as hyperfine coupling will boost our atomistic understanding of the underlying mechanisms.'

We claim to have molecule-to-molecule resolution in ESR-AFM, as we demonstrate in our manuscript. We state that we combine the high energy resolution of the ESR-AFM with atomic-scale spatial resolution, but don't imply intra-molecular spatial resolution of ESR-AFM itself.

Actions:

We now discuss the spatial resolution of ESR-AFM directly in the main text. A discussion of restrictions concerning tip height we included in the methods section and refer to that in the main text.

We understand that the phrase "combines sub-nanoelectronvolt spectral and atomic-scale spatial resolution" could be misunderstood as a claim of intramolecular resolution of ESR-AFM itself and therefore removed it.

Due to length restrictions we also had to shorten the title. We believe that the new title is also less controversial with respect to the points raised by the referee.

(ii) They give zero discussion about what they can learn about how the environment changes the coherent behavior (e.g. do they see a difference between coherence times for different molecules, due to defects). This is the main claim of their paper, yet the paper only exhaustively wants to claim that they are studying a nearly free standing molecule and their method does not perturb the measurement. They want to claim there are no defects (which is not believable and even looks present in Fig. 2C), and that their method does not perturb the molecule. Ionic defects and step edges in their insulator can present significant potential variations that can change their level structure. Then they see a variance in their statistics, which they do not explain and need to account for (textbook error analysis). Why is there a change in their frequencies in extended table 1? There's no change between metal and CO tips, where the latter may even have a strong electric dipole. Does this mean that they are not sensitive to the atomic environment, which then a contradiction to what they claim? I would guess that the points "change the quantum system" as the environment is changed. If there are no effects there, then they cannot claim their method is sensitive to any of this, and thus it is contradictory. This needs to be discussed and clarified, as it is very confusing.

Authors: *We thank the reviewer for pointing out that this remained unclear. We believe that the confusion arises from the following: The different influences on the spin system occur on very different magnitudes. Our claim is that by our approach, we eliminate many of the decoherence sources that **strongly** perturb the system. Thereby, one has access to all those perturbations that are much weaker.*

*We never wanted to claim that we do not perturb the system at all, which we also clarified in our last revision. Providing some numbers might help clarifying this point: Previous local-probe ESR measurements demonstrated line widths on the order of a few MHz; here we demonstrate a line broadening more than one order of magnitude sharper. We do observe a Stark shift from the electric field in the junction (where the overall electric field seems to be dominating in comparison to the difference of metal versus CO tips at the given tip-sample distance). The largest possible variation of field accessible in our experiment for pentacene yielded a shift of the line of only ~ 0.3 MHz. A hydrogen nuclear spin inside pentacene typically couples to the spin system on the scale of below 0.5 mT (see e.g., *Electron Spin Resonance Spectroscopy of Organic Radicals*. Gerson and Huber, Wiley-VCH, ISBN: 3-527-30275-1). In absence of an external magnetic field, this shifts the transitions by up to a few MHz. We do expect nuclear spins in the substrate to couple to the molecular system, but on a substantially smaller scale. As the referee alludes to, we do see small shifts of the ESR signal depending how close a molecule lies to a step edge of NaCl. A detailed analysis of such an effect goes beyond the scope of the current manuscript, but we now mention this in the caption of Extended Data Table 1.*

We refer to such effects in the very last sentence of the abstract, which – according to nature's style guide – puts the work into future perspective. To emphasize that even more we changed this sentence, now starting with "We anticipate that..."

The referee probably recognized that the observed variations are on the order of 1 MHz, relating to less than 0.1% relative variations. With decoherence and broadening present on the scale of previous local-probe ESR experiments, some of these variations would hardly be recognizable in the first place.

Hence, the point that we indeed want to make is that many of the large sources of coupling and decoherences are eliminated, now opening the door for more subtle effects to be studied in the future. No doubt, these more subtle effects are there, and we do already get a glimpse of them. We try to implement this aspect better in the revised version, but if the referee gave us a specific hint, how to further improve on that, we will be happy to do so.

Actions:

We completely revised the discussion of the strong and weak perturbations in the spirit of the above discussion. We explicitly mention defects as step edges in the NaCl.

2. R1

In the discussion about energy resolution, the authors do not address the underlying issue about what their energy resolution is, or if they can claim it. This was linked to the discussion about thermodynamics and non-equilibrium. They provide a lengthy and confusing discussion, some of which I believe is not completely correct, about thermodynamics. The original point is the word energy resolution cannot be used in the context they describe. To claim this is an energy resolution of the method, it must be determined against a known benchmark, and arguably done in equilibrium. This would illustrate that the energy resolution is inherent to the method. For example: is the energy resolution dependent on the molecule they probe, or the dynamics they excite? To be clear, temperature in this system should change intensities of these peaks, and one cannot compare this method with the ensemble measurements they try to argue with (e.g. where bosons are also studied). Here, the question is if what they claim as resolution is dependent on the molecule and their excitation conditions.

Authors: *Unfortunately, we do not fully grasp the reviewer's irritation about our wording. We therefore try to answer concerning different aspects:*

(a) We believe "energy resolution" is the right term in the context of our specific experiment: By probing the ESR spectrum of the T_X - T_Z transition, we in fact determine the energy splitting between the T_X and the T_Z state. Using pentacene as a known benchmark together with existing literature strongly supports that we indeed probe an energy difference.

Once this is settled, the only scaling factor between this energy difference and the observed resonance frequency in the ESR signal is the Planck constant. Hence our energy resolution is directly tied to the precision in determining the resonance frequency in the ESR signal. From our spectra we could in principle read off the frequency at the maximum even more precisely than the claimed energy resolution, but we follow the usual convention to take the FWHM of the sharpest feature that we can resolve as the energy resolution.

We note that, the uncertainty in the frequency selection as well as the uncertainty in the Planck constant are negligible on the scale of our claimed energy resolution.

(b) Maybe the referee wants to express that the energy resolution of our specific experiment is not necessarily "the" energy resolution of the method in general.

We fully agree with that. In this sense, the energy resolution demonstrated here can only represent an upper bound to the possible energy resolution of the method in general. This relates to the referee's point that the resolution might change from molecule to molecule and, for example, depends on the excitation conditions. It is true that, too strong excitation results in power broadening that is associated to a loss in energy resolution, as we demonstrated in Extended Data Fig. 4. But since all this can only impair the energy resolution (and not make it artificially appear better than it actually is), the energy resolution demonstrated here should be considered as an upper bound to the intrinsic energy resolution of the method. We understand that our previous phrasing was not entirely clear in that respect.

Actions:

Hoping that this addresses the referee's concern, we now mention the energy equivalent of our spectral features (sub-nanoelectronvolt) only in direct relation to our specific experiment and ESR feature and not as a general property of ESR-AFM. We further changed all occurrences of "energy resolution" to "spectral resolution".

3. R1

Often, the authors use words like isolated as they want to accentuate that they are not strongly coupled to an electron bath. But, this reads very contradictory in their paper. Isolated quantum systems are typically called closed quantum systems, and they are far from this limit. Also, even though they suppress electron scattering does not mean they can ignore decoherence from other sources (e.g. phonons, dynamically induced distortions of the

molecule, tip/molecule interactions, other electric fields generated by defects, etc.). What is the role of Brownian motion of the tip? There's no satisfactory discussion about the limits of the method, and what sources of decoherence exist. Simply statements about what does not affect the measurement. In the tone of the manuscript, and the poor comparison to ESR-STM, they signal that they are non-invasively probing a closed quantum system.

Authors:

We understand very well that our very first version of the manuscript provided the false impression that we only excluded possible effects of decoherence. In the revised version we tried to improve on this aspect, by naming two specific sources of decoherence (hyperfine interaction and scattering with electrons in the tip).

In view of the referee's comment and the previous repeated criticism, we now wrote an explicit paragraph that introduces multiple possible decoherence sources (after discussing which sources of decoherence are absent). For example, we now explicitly mention the Stark effect and the motion of the cantilever (since the RF pulse was on during multiple cantilever cycles Brownian motion is negligible against the regular cantilever movement, which needs to be taken into account).

We would like to include a discussion discriminating between static interactions that can shift the resonances, as for example, a static electric field from a defect, and perturbations that are fluctuating in time, which lead to decoherence. However, already such a basic discussion seems to exceed the limited space we have.

We note, however, that not all limitation of this technique might be known so far, and that, historically, hardly any previous introductions of novel methods have named all their limitations, and, finally, that manuscript space is quite limited. We kindly ask, therefore, the referee to sympathetically judge on our revision baring the above in mind.

Already in our last revision we substantially reduced the extend and emphasis of the comparison to ESR-STM. We now removed the one remaining direct comparison to ESR-STM such that there is no explicit comparison to ESR-STM left.

Actions:

We now explicitly name a number of possible decoherence sources. We mention the Stark effect and that it may considerably contribute to broadening. We mention the effect of vertical tip motion. We deleted without replacement the half sentence "likely not limited by the detection method" as well as the entire sentence "Further, it seems likely that for deuterated pentacene the coherence time is still limited by the smaller but nonzero interaction with the nuclear spins and the resulting fluctuating Rabi frequency." We removed all explicit comparisons to ESR-STM.

4. R1

If it is correct that PTCDA was measured twice, it needs to be removed from the manuscript, or more properly studied. To standards in the field, this is not considered statistics and should not be published, unless it is clearly discussed, including the lack of statistics. If they want to include this molecule in the paper, they really need to better describe what they measure beyond one sentence.

Authors: *PTCDA has been measured more than twice. In fact, we could even observe all three transitions, but we show only the strongest in the manuscript. Alone in the manuscript we present two spectra: a large scale and a close up. In the course of our experiments we acquired 28 spectra on PTCDA. We now included this in the statistical information that we present in the methods section.*

We regret that the length restrictions do not allow us to expand the discussion of the PTCDA results in the main text. PTCDA was included upon request of the other referee to demonstrate the broader applicability. We believe to have served this purpose. The details of the PTCDA experiments are mentioned in the methods (which may have been overlooked), with the different aspects in the contextually corresponding paragraphs.

Actions:

We extended the statistical information for PTCDA, including the number of spectra we took.

Minor points:

5. R1

I found many instances of the word “tiny” in the manuscript, which is not scientific language.

Authors: *We thank the reviewer for noting this.*

Actions:

We replaced all instances of tiny in the manuscript.

6. R1

I think the introduction is overstated, for example “the construction of understanding of decoherence in complex solid-state quantum devices requires ultimately controlling the environment down to atomic scales...” A large part of the solid-state community in quantum technology focuses on superconductors. The length scale of interest is the coherence length, which is on the order of 10s of nms, and not focused on atomic-scale control.”

Authors: *We thank the reviewer for making us aware that the length scale of interest is not for every community in quantum technology on the atomic scale.*

Actions:

We toned down our sentence by adding the word “many” before complex solid-state quantum devices. To avoid the false impression that we aim at direct technological applications, we removed all instances of “devices” in the context of our work. We added “fundamental” to clarify, in which context we see possible applications of ESR-AFM for quantum sensing, namely, with respect to fundamental science.

7. R1

I would refrain from using words like isolated in context of quantum systems, as reasoned above.

Actions:

We removed the word isolated from our manuscript.

Referee #2 (Remarks to the Author)

In the 2023 resubmission of "Atomic-scale access to long spin coherence in single molecules with scanning force microscopy" the authors have responded in detail to the comments by both referees. They have made many changes and included major new data that greatly strengthen the paper. The inclusion of the PTCDA molecule, the wide-frequency spectra of both molecules, and the clear correspondence between bond-resolved images and their ESR spectra, all acquired using the same tip, are an outstanding experimental accomplishment that supports the main claims. Other new information improves the paper markedly, such as the Rabi dependence on the drive amplitude, the acquisition statistics, the line shape explanation, the selection rules, and the discussion of Rabi mechanisms, rates, and molecular orientation. These changes strengthen the paper to the level that I recommend publication in Nature.

Authors: *We thank the referee for their appreciation of the data and information we newly included and are thankful for their comments to further improve our manuscript, which will be addressed below.*

1. R2

A plausible decoherence mechanism that was not mentioned earlier is the fluctuating Stark shift of the resonant frequency that is due to the uncontrolled electric field fluctuations that arise from uncontrolled motion of the conductive AFM tip. Please pardon my oversight in neglecting to discuss Stark shifts in the first evaluation. Decoherence is emphasized in this paper, so I recommend mentioning the Stark shift mechanism and providing an estimate of the range of plausible values for the line broadening, even if this discussion simply rules out observable Stark effects here. A rough estimate: at a bias voltage of 2.5 V during the dwell period, an effective junction thickness of ~2.5 nm gives an electric field of ~1 GV/m. If the unintended fluctuating tip motion is ~2.5 pm, then the electric field fluctuations are ~1 MV/m. A Stark shift of 1 Hz/(V/m) would give an ESR line broadening of ~1 MHz, which is in the range of the broadening observed. Stark shift sensitivities vary widely, and the authors may have better estimates of all these parameters given their specific spin system and experimental setup. The tunnel barrier includes the thick NaCl, but this has a large dielectric constant, so its vacuum-equivalent effective thickness is less. A determination of this Stark-shift sensitivity may already be available to the authors if they measured how much the frequency shifts with different dwell voltages or with significantly different NaCl thicknesses. Extended Fig. 7 is a step in that direction but doesn't vary V or vary z enough to estimate the effect. An indication that Stark shift may be negligible is the observation that the cantilever oscillation (0.1 nm zero-to-peak or peak-to-peak?) occurs on the same timescale as the Rabi RF pulse duration but there does not seem to be a dependence on the cantilever phase relative to the RF pulse. Do the authors synchronize the start of the RF pulse with the cantilever as they do for the charge state control, or is this unnecessary?

Authors: *We thank the reviewer for raising this point and the elaborate discussion. We fully agree that we should mention the Stark shift mechanism as a source of line broadening. As we now explicitly indicate in the methods, the stated amplitude is the zero-to-peak amplitude. (In the course of these considerations we realized that our amplitude calibration was off, which is now corrected for). While it is not necessary for the experiment, in the case of the ESR-AFM spectra, the RF pulse was indirectly synchronized with the cantilever movement, since it occurs with a fixed delay w.r.t. the dwell voltage pulse, which in turn is synchronized with the cantilever movement. Important to note here: the RF pulse was on during multiple cantilever cycles. Hence, undesired tip fluctuations are negligible against the regular cantilever movement, which needs to be taken into account. Irrespective of the referee's comment, we were also concerned with the Stark shift. In fact, we hope to even exploit the Stark shift in the future for scanning-gate-type experiments. In this context, in the meantime we performed experiments, in which we observed a shift of the*

ESR signal of ~ 0.3 MHz per change of sample voltage by 1 V. This Stark shift is not at all negligible. Hence, the regular tip height oscillation may lead to a periodic shift of the ESR line and thereby significantly contribute to the observed finite line width, which we now clearly state in the manuscript.

We thank the referee again for bringing this up and fully agree with the referee that this should be discussed.

Action: We included a discussion of the Stark shift and its consequences into the manuscript.

The following may optionally help to improve the clarity:

2. R2

Line 135: In the new paragraph “The corresponding Rabi oscillations...”: corresponding to what? Presumably this means deuterated pentacene.

Authors: We thank the reviewer for making us aware that this phrase was not clear. We indeed meant deuterated pentacene.

Actions:

We improved this sentence.

3. R2

Line 158: How is being “out-of-thermal-equilibrium” relevant to measuring at the conditions used? Does it mean that the initialized state is other than Boltzmann, or does it mean something else?

Authors: Out-of-thermal-equilibrium is relevant since we are not performing ESR measurements of the ground state of our molecule, but of the triplet excited state. The imbalance in populations in the triplet substates (after initializing them with equal populations) arises from their different decay constants back to the ground state, and thereby strongly deviates from a Boltzmann distribution. Importantly, this will in first approximation be independent of temperature, allowing us to work at $T = 8$ K, for which $k_B T \gg \Delta E$ of the zero-field splitting.

4. R2

Line 161: “The increased spin coherence...”: rephrase or state what it’s compared to.

Authors: We thank the reviewer for noting that this was not phrased clearly.

Actions:

We rephrased this sentence.

5. R2

Line 282: “represented by the two arrows” perhaps refers to the tiny black arrows on the molecule added in this version. These arrows may represent spins but are illegible at this scale and are not needed.

Authors: We thank the reviewer for noting that the spins were drawn too small; as it was requested by the other reviewer to include these spins, we decided to keep them but increase their size to improve readability.

Actions:

We made the indicated suggestion to figure 1a.

6. R2

Line 286: “Decay of T1”: The authors are careful to distinguish the label T1 (any of the triplet states) from the variable tau (lifetimes), but T1 is used for relaxation times in many other publications, so this may confuse casual readers. Consider adding a few clarifying words like

“decay of the triplet states”; and “the lifetime measurements of the triplet state T1” instead of “the T1 lifetime measurements”.

Authors: *We fully agree with the reviewer that this might cause confusion, especially in the cases like those cited.*

Actions:

We implemented the suggestions of the reviewer.

7. R2

Line 299: “perdeuterated pentacene-d14 (A) and PTCDA (B)” is unclear whether it is perdeuterated PTCDA.

Authors: *We thank the reviewer for noting our confusing notation here. Our PTCDA is not perdeuterated.*

Actions:

To avoid confusing, we removed the word perdeuterated in this line, and made explicit in the methods section that the used PTCDA was protonated.

8. R2

Line 300: “Note that for PTCDA there is a tiny signal at 1501 MHz that is too small to be observed at the parameters used here”: Did the authors observe this signal at some other parameters? If it is not shown anywhere in this paper, say something like “not shown”.

Authors: *We did indeed observe this signal at some other parameters.*

Actions:

We added “not shown” to this line.

9. R2

Line 317: The variable t_S is used before being defined, and what does the letter “s” stand for? The 45.1 microsecond delay mentioned in line 97 is presumably t_S . The times t_S and t_D are illustrated in the inset of Fig. 3a but are too small to be legible; taking space for that diagram is worth it.

Authors: *We thank the reviewer for their careful observation that we did not define t_S before using it in the figure caption. The letter S stands for the Start of the RF pulse, as we also indicate in a few places in the methods section.*

We also thank the reviewer for suggestion to make the inset of Fig. 3a larger. Unfortunately, bound by the font sizes of Nature style we cannot make the font sizes of t_S and t_D larger. We trust in the Nature copy-editing department to solve the problem during the copy-editing process.

Actions:

We introduce the abbreviation t_S now in line 97.

10. R2

Line 319 “...as indicated”: indicated where?

Authors: *This is indicated for the first oscillation in figure 3a.*

Actions:

We now explicitly state this.

11. R2

Extended Data Fig. 1: The yellow labels P+ and D0 are too light to be legible.

Authors: *We thank the reviewer for making us aware of this.*

Actions:

We changed the colors as requested.

12. R2

Line 801: “two per orientation”: are there precisely two orientations observed?

Authors: *On the NaCl island that we are probing there are indeed precisely two molecular orientations observed, due to the two possible adsorption geometries of pentacene.*

Actions:

The orientation of pentacene is now introduced in the Methods section with a corresponding reference.

Reviewer Reports on the Second Revision:

Referees' comments:

Referee #1 (Remarks to the Author):

In this manuscript revision, the authors have greatly improved the readability of the manuscript. I would like to thank the authors for their openness and discussion, and congratulate them on the quality of these results. I do believe that the observation of ESR signal on a second molecule and the expanded discussion of potential decoherence, will lead to many exciting directions in quantum sensing. I also believe the quality of the manuscript has been greatly improved. I believe this can be published as is, pending one very minor suggestion. On page 5, where there is a description of the z-range used to detect, the authors should add a reference or sentence to the x-y resolution discussion (e.g. Extended data 5). They have addressed this in the paper. However, it would read better if there was some link to it in this section, as the z-range is also discussed.

Author Rebuttals to Second Revision:

Referee #1 (Remarks to the Author):

In this manuscript revision, the authors have greatly improved the readability of the manuscript. I would like to thank the authors for their openness and discussion, and congratulate them on the quality of these results. I do believe that the observation of ESR signal on a second molecule and the expanded discussion of potential decoherence, will lead to many exciting directions in quantum sensing. I also believe the quality of the manuscript has been greatly improved. I believe this can be published as is, pending one very minor suggestion. On page 5, where there is a description of the z-range used to detect, the authors should add a reference or sentence to the x-y resolution discussion (e.g. Extended data 5). They have addressed this in the paper. However, it would read better if there was some link to it in this section, as the z-range is also discussed.

Authors: *We are delighted that we could convince the referee of the quality of our work and that they now recommend this version for publication. We thank the referee for their intense work in the entire review process, for their patience, and for their positive remarks.*

We fully agree with the final suggestion of the referee.

Actions:

From the main text at the spot mentioned by the referee, there existed already a reference to both, the Methods and Extended Data Fig. 5. As suggested, we now inserted a reverse link from the respective text section in the Methods to the main text as “see also discussion of the spatial resolution in the main text”. A similar reference to the main text is now introduced into the caption of Extended Data Fig. 5.